# UNDERWATER VISUAL GEOMETRY ESTIMATION WITH SELF-SUPERVISED PROTOTYPE-GRAPH MODULATION

## ABSTRACT

Underwater 3D reconstruction poses significant challenges due to the scarcity of large-scale labeled datasets and the lack of foundation models specifically designed for underwater scenarios. To overcome these limitations, we introduce **SeaVGGT**, a self-supervised framework for underwater geometric estimation that operates without reliance on annotated data or enhancement references. SeaVGGT exploits the fundamental physical principle that underwater image degradation inherently encodes scene depth, and captures this phenomenon through a graph of learnable prototypes. These prototypes encapsulate a diverse range of attenuation characteristics and are dynamically selected as context-aware conditions to modulate visual features in a depth-sensitive manner. The framework is trained in an end-to-end fashion using a set of physics-driven self-supervision losses, which enforces cyclic consistency between the original and reconstructed images based on the underwater imaging formation model. To robustly handle the variability of water types and environmental conditions, SeaVGGT adaptively refines prototype representations conditioned on the input image, thereby enabling strong generalization across diverse underwater domains. Extensive experiments on FLSea, USOD10K, and SQUID datasets demonstrate that SeaVGGT achieves a 13.47% reduction in RMSE under unseen water conditions compared to the VGGT baseline, underscoring its efficacy and broad applicability.

## 1 INTRODUCTION

Accurate depth estimation in underwater environments is essential for a wide range of applications, including marine robotics, ecological monitoring, and archaeological site exploration. Unlike terrestrial scenes, underwater imaging Lee et al. (2023); Qi et al. (2025); Zhang et al. (2024a); Chang et al. (2025); Cong et al. (2024); Guo et al. (2023); Naik et al. (2021) suffers from complex optical degradations such as wavelength-dependent light attenuation, scattering by suspended particles, and severe color distortions. These effects significantly degrade the visual cues that conventional depth estimation models rely on, making underwater depth prediction a uniquely challenging problem.

Moreover, acquiring large-scale, densely annotated underwater depth datasets is technically challenging and economically prohibitive. First, common RGB-D sensors such as structured-light or time-of-flight cameras degrade significantly underwater due to limited effective range and severe distortions, while sonar sensors produce sparse, low-resolution depth maps that are unsuitable for training deep learning models. Second, state-of-the-art foundational depth estimation models pretrained on terrestrial datasets, such as DepthAnything Yang et al. (2024b), MASt3R Leroy et al. (2024), and VGGT Wang et al. (2025), suffer from substantial performance degradation when directly applied to underwater imagery because of domain shifts caused by wavelength-dependent attenuation and scattering. Third, retraining or fine-tuning these models with limited underwater labeled data is challenging due to the scarcity of annotations and the risk of overfitting. These challenges motivate the development of label-efficient adaptation methods that leverage pretrained terrestrial models while effectively bridging the domain gap to enable robust underwater depth estimation.

In this work, we propose **SeaVGGT**, a self-supervised framework designed to adapt pretrained depth models to underwater environments without requiring any ground truth annotations or enhancement

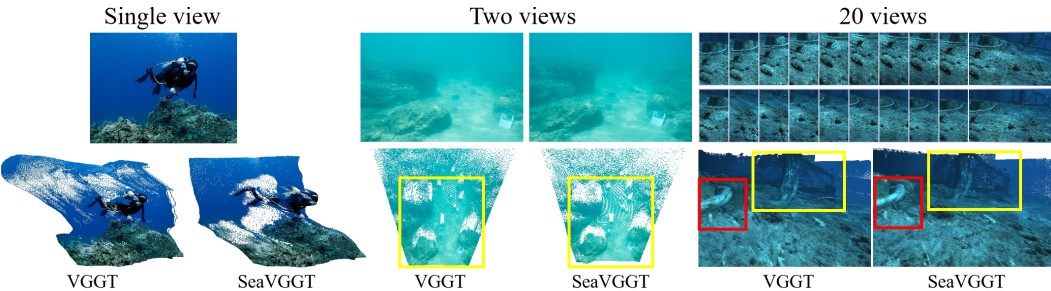

Figure 1: **SeaVGGT**, an efficient adaptation of the original VGGT tailored for underwater scenes, achieves remarkable improvements in 3D structure reconstruction, boundary clarity, and multi-view consistency with minimal additional computational cost and label requirements. Leveraging a self-supervised learning paradigm, *SeaVGGT does not rely on annotated data and achieves efficient adaptation without any fine-tuning of the VGGT backbone or prediction heads*, making it highly practical for diverse underwater environments.

references. Our method explicitly models underwater image degradation as structured biases in latent feature space, characterized by a set of learnable prototypes that represent diverse water types and attenuation patterns. These prototypes are connected in a graph structure, enabling relational reasoning through message passing that captures complex dependencies between different water conditions.

SeaVGGT employs a lightweight token modulation mechanism conditioned on context-aware prototype representations, facilitating depth-relevant feature adaptation while preserving the structural priors of the underlying pretrained model, as shown in Figure 1. The entire system is trained end-to-end with a physics-driven self-supervision loss, grounded in the underwater image formation model, enforcing cyclic consistency between input images and their reconstructions. To handle variability and uncertainty in water types encountered during inference, SeaVGGT further introduces a self-evolving prototype graph that dynamically refines prototype representations based on input image statistics. Our main contributions are:

- A prototype-guided token modulation framework that adapts a pretrained VGGT model to diverse underwater conditions by modeling complex degradation patterns without requiring labeled data.
- A graph-based prototype learning strategy combined with a physics-driven self-supervised loss, enabling relational modeling among water types and accurate geometric estimation based on underwater image formation principles.

## 2 RELATED WORK

**Underwater Scene Geometry Understanding.** Early methods focused on image enhancement to restore underwater images using physical models of light attenuation and scattering Zhou et al. (2023), yet struggled in turbid or variable conditions. Recent supervised deep learning approaches estimate depth or 3D structure directly from monocular underwater images Hambarde et al. (2021), but rely on scarce annotated data. Synthetic datasets Zhao et al. (2021); Zwilgmeyer et al. (2021) partially alleviate this but suffer from domain gaps. Transfer learning from terrestrial pretrained models with fine-tuning on limited underwater data Yu et al. (2023); Ebner et al. (2024) has shown promise, using architectures like Monodepth2 Godard et al. (2019) and AdaBins Bhat et al. (2021), yet still requires supervision and may reduce generality. Beyond monocular depth, comprehensive underwater geometry tasks remain underexplored due to data scarcity and optical complexities. Our work proposes a self-supervised framework that jointly models underwater degradation and geometry without ground truth, enabling robust scene understanding across diverse conditions.

**Underwater Image Enhancement.** Traditional physical model-based underwater image enhancement (UIE) methods Drews Jr et al. (2013); Wang et al. (2018); Chiang & Chen (2012); Li et al. (2016); Akkaynak & Treibitz (2019) use hand-crafted priors and parameter estimation to invert degradation, but depth recovery remains challenging and limits robustness. CNN-based UIE

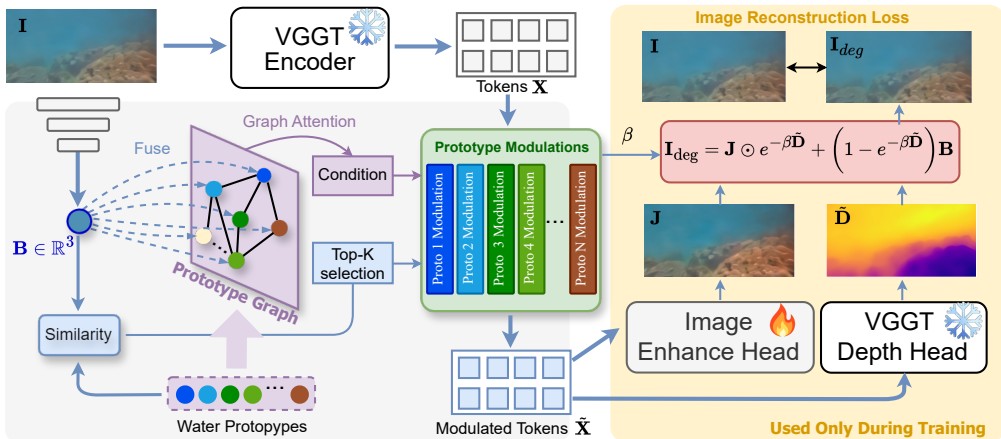

Figure 2: Overview of SeaVGGT. Given an underwater image, visual tokens are extracted via VGGT's encoder and modulated by water-type-aware prototypes to adapt the features to diverse underwater conditions. The prototype representations are refined via graph attention and selectively integrated to guide the modulation. The entire system is trained under a self-supervised learning paradigm leveraging physics-based constraints, enabling effective adaptation without reliance on annotated data. The resulting modulated token features support downstream tasks such as depth estimation and camera pose prediction.

approaches Li et al. (2017); Fabbri et al. (2018); Uplavikar et al. (2019); Li et al. (2021); Zhou et al. (2024); Li et al. (2019) employ end-to-end or parameter estimation networks, improving speed and generality but relying on paired or synthetic data that may not fully represent real underwater conditions, restricting geometric understanding. Transformer-based UIE methods Dosovitskiy et al. (2020); Liu et al. (2021); Peng et al. (2023); Tang et al. (2023); Khan et al. (2024); Peng & Bian (2025) capture global context yet face computational costs and uneven degradation challenges; recent modules adaptively model degradation variation to enhance regional image quality. Such enhancement techniques increasingly serve not only preprocessing but also as supervision signals for self-supervised underwater geometry estimation without ground truth.

## 3 METHOD

Underwater images exhibit complex variations due to the intertwined effects of depth and color distortion, posing challenges for generalizable depth estimation. To tackle this, we propose a water-prototype-aware modulation framework that leverages learned water-type prototypes to adaptively modulate token features under a self-supervised learning paradigm guided by underwater imaging physics. Figure 2 provides an overview of the architecture.

### 3.1 PROBLEM FORMULATION AND KEY CHALLENGES

We first revisit the formulation of VGGT, which serves as the foundation of our approach. Given a set of $N$ RGB images $\{\mathbf{I}_i\}_{i=1}^{N}$ captured from different viewpoints of the same scene, VGGT aims to predict, for each image $\mathbf{I}_i$, a per-pixel depth map $\mathbf{D}_i \in \mathbb{R}^{H \times W}$, a point cloud map $\mathbf{Q}_i \in \mathbb{R}^{H \times W \times 3}$ in the world coordinate system, and camera parameters, including the intrinsic matrix $\mathbf{K}_i \in \mathbb{R}^{3 \times 3}$ and extrinsic matrix $\mathbf{T}_i \in \mathbb{R}^{4 \times 4}$.

To this end, each image $\mathbf{I}_i$ is tokenized into patch tokens, which are then jointly processed by a set of frame-wise and global self-attention layers with cross-frame interactions. The resulting $K$ feature tokens $\mathbf{X}_i = \{\mathbf{x}_{i,k} \in \mathbb{R}^C\}_{k=1}^{K}$ are passed through task-specific heads to infer the desired geometric and camera outputs:

$$\mathbf{D}_i = f_D(\mathbf{X}_i), \quad \mathbf{Q}_i = f_Q(\mathbf{X}_i), \quad (\mathbf{K}_i, \mathbf{T}_i) = f_{\text{cam}}(\mathbf{X}_i), \tag{1}$$

where $f_D$, $f_Q$, and $f_{\text{cam}}$ denote the prediction heads for depth, point cloud, and camera parameters, respectively.

To facilitate the robust application of VGGT in underwater environments, which present significant domain shifts and limited annotated data, we introduce a lightweight token modulation mechanism.

This mechanism adaptively refines the extracted feature tokens $\mathbf{X}_i$ to capture underwater-specific image characteristics, yielding modulated tokens $\tilde{\mathbf{X}}_i$ that preserve the original feature dimensionality and spatial organization. Crucially, these modulated tokens can be directly processed by VGGT's pretrained prediction heads:

$$\tilde{\mathbf{D}}_i = f_D(\tilde{\mathbf{X}}_i), \quad \tilde{\mathbf{Q}}_i = f_Q(\tilde{\mathbf{X}}_i), \quad (\tilde{\mathbf{K}}_i, \tilde{\mathbf{T}}_i) = f_{\text{cam}}(\tilde{\mathbf{X}}_i). \tag{2}$$

This design leverages the strong pretrained geometric reasoning of VGGT while enabling flexible adaptation to the unique visual properties of underwater scenes. By performing modulation at the token level, our method obviates the need for costly retraining of the backbone or prediction heads, thereby substantially reducing computational overhead and annotation requirements. Such an approach facilitates efficient and practical deployment of VGGT in real-world underwater applications.

## 3.2 PHYSICS-GUIDED SELF-SUPERVISION FRAMEWORK

We propose a physics-guided self-supervision framework that jointly predicts a restored image $\mathbf{J}$ and a corresponding depth map $\mathbf{D}$ from the modulated feature $\tilde{\mathbf{X}}$. The framework consists of two branches: an image enhancement head $f_J$ and a depth estimation head $f_D$, where the former is randomly initialized and trained from scratch, while the latter is from the pre-trained VGGT model with fixed parameters. We also introduce a learnable 3-dimensional vector $\beta$ that models the attenuation coefficient in underwater imaging. The degraded image $\mathbf{I}_{\text{deg}}$ is reconstructed via a simplified underwater imaging model:

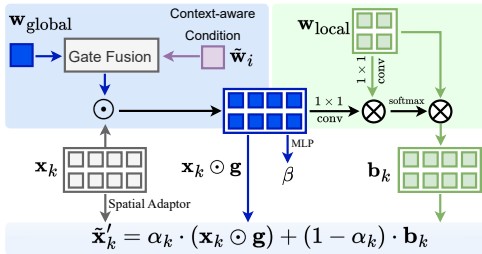

Figure 3: Illustration of prototype-guided token modulation.

$$\mathbf{I}_{\text{deg}} = \mathbf{J} \odot e^{-\beta \tilde{\mathbf{D}}} + \left(1 - e^{-\beta \tilde{\mathbf{D}}}\right) \mathbf{B} = f_J(\tilde{\mathbf{X}}) \odot e^{-\beta f_D(\tilde{\mathbf{X}})} + \left(1 - e^{-\beta f_D(\tilde{\mathbf{X}})}\right) \mathbf{B}, \tag{3}$$

where $\mathbf{B}$ denotes the background light, assumed to be constant and estimated from the input image $\mathbf{I}$ via a 3-layer convolutional neural network $f_B$, as illustrated in the upper-left part of Figure 2.

To regularize the self-supervised learning, we define the following self-reconstruction losses:

$$\begin{aligned} \mathcal{L}_{\text{rec}} = &\lambda_{l1} \|\mathbf{I}_{\text{deg}} - \mathbf{I}\|_1 + \lambda_{\text{ssim}} \left(1 - \text{SSIM}(\mathbf{I}_{\text{deg}}, \mathbf{I})\right) \\ &+ \lambda_{\text{percep}} \|\phi(\mathbf{I}_{\text{deg}}) - \phi(\mathbf{I})\|_1 \\ &+ \lambda_{\text{grad}} \left(\|\nabla \mathbf{I}_{\text{deg}_x} - \nabla \mathbf{I}_x\|_1 + \left\|\nabla \mathbf{I}_{\text{deg}_y} - \nabla \mathbf{I}_y\right\|_1\right), \end{aligned} \tag{4}$$

where $\phi(\cdot)$ extracts perceptual features from a fixed VGG16 pretrained network, and $\nabla \mathbf{I}_x$, $\nabla \mathbf{I}_y$ denote spatial gradients along width and height directions.

Inspired by USUIR Fu et al. (2022), the enhanced image $\mathbf{J}$ is constrained to be consistent with a mixup-augmented version $\mathbf{J}_{mix}$ via a mean squared error loss:

$$\mathcal{L}_{\text{enh}} = \|\mathbf{J}_{\text{mix}} - \mathbf{J}\|_2^2. \tag{5}$$

To encourage neutral color tones in $\mathbf{J}$, we penalize deviation of channel-wise means from $0.5$:

$$\mathcal{L}_{\text{color}} = \sqrt{(m_r - 0.5)^4 + (m_g - 0.5)^4 + (m_b - 0.5)^4}, \tag{6}$$

where $m_c$ is the mean intensity of channel $c \in \{r, g, b\}$.

Since the farthest pixels in underwater scenes are minimally affected by reflected object color, the background color $\mathbf{B} \in \mathbb{R}^3$ is supervised by a proxy ground truth. Specifically, for each image, we select the top $0.1\%$ of pixels with the largest depth values and compute the average RGB value at those locations:

$$\mathcal{L}_{\text{bg}} = \left\|\mathbf{B} - \frac{1}{|\mathcal{P}|} \sum_{(i,j) \in \mathcal{P}} \mathbf{I}(i,j,:)\right\|_2^2, \tag{7}$$

where $\mathcal{P} = \text{Top}-0.1\%\,(\mathbf{D})$ denotes the set of pixel locations with the top $0.1\%$ largest depth values in the image. To mitigate potential inaccuracies of $\mathbf{D}$ under domain shift, we assign this term only a small weight, ensuring that it serves as a weak regularizer rather than a dominant supervision signal. Therefore, the overall loss of the framework is:

$$\mathcal{L}_{\text{total}} = \lambda_{\text{rec}}\mathcal{L}_{\text{rec}} + \lambda_{\text{enh}}\mathcal{L}_{\text{enh}} + \lambda_{\text{color}}\mathcal{L}_{\text{color}} + \lambda_{\text{bg}}\mathcal{L}_{\text{bg}}, \tag{8}$$

where the coefficients $\lambda_{\text{rec}}$, $\lambda_{\text{enh}}$, $\lambda_{\text{color}}$, and $\lambda_{\text{bg}}$ are loss weights, which are specified in the experimental details.

## 3.3 WATER-PROTOTYPE GRAPH CONSTRUCTION

To model appearance variations across water types, we construct a water-prototype graph $\mathcal{G} = (\mathcal{V}, \mathcal{E})$, where each node represents a learnable prototype $\mathbf{w}_i$ associated with an RGB color vector $\mathbf{A}_i \in \mathbb{R}^3$ indicating typical underwater color and scattering behavior. The color matrix $\mathbf{A} \in \mathbb{R}^{P \times 3}$ is initialized from prior waterbody samples and optimized during training.

**Graph Edges.** Edges $\mathcal{E}$ are established between visually similar prototypes based on color distance:

$$d_{ij} = \|\mathbf{A}_i - \mathbf{A}_j\|_2, \quad \mathbf{E}_{\text{adj}}[i, j] = \mathbb{I}[d_{ij} < \tau], \tag{9}$$

with threshold $\tau = 0.3$. This restricts message passing to perceptually similar water types.

**GAT Input.** Each node $\mathbf{w}_i$ is further conditioned on the predicted global background color $\mathbf{B} \in \mathbb{R}^3$. We concatenate $\mathbf{A}_i$ and $\mathbf{B}$ to form the input to the graph attention network:

$$\mathbf{H}_i = [\mathbf{A}_i \,\|\, \mathbf{B}] \in \mathbb{R}^6, \tag{10}$$

resulting in the full input matrix $\mathbf{H} \in \mathbb{R}^{P \times 6}$. This concatenation allows the GAT to refine each prototype in a context-aware manner, adapting node features according to how relevant each water type is to the current input image.

**Prototype Feature Update via GAT.** Each prototype node is refined using a single-layer graph attention, which incorporates context from visually similar water types. Given the input feature matrix $\mathbf{H} \in \mathbb{R}^{P \times 6}$, each node is first linearly projected to a hidden space with learnable parameters $\mathbf{W}$:

$$\mathbf{Z}_i = \mathbf{W}\mathbf{H}_i, \quad \mathbf{W} \in \mathbb{R}^{6 \times d}, \tag{11}$$

and the attention coefficients between node $i$ and $j$ are computed as:

$$e_{ij} = \begin{cases} \phi\left(\mathbf{a}^\top [\mathbf{Z}_i \,\|\, \mathbf{Z}_j]\right), & \text{if } (i, j) \in \mathcal{E}, \\ -\infty, & \text{otherwise}, \end{cases} \tag{12}$$

where $\mathbf{a} \in \mathbb{R}^{2d}$ is a learnable parameter and $\phi(\cdot)$ denotes the LeakyReLU activation.

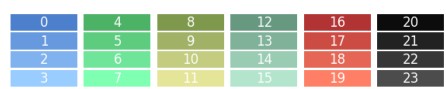

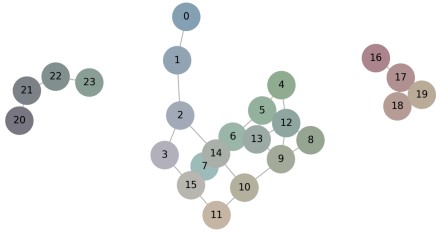

(a) Initial colors of water exemplars

(b) Water-exemplar graph after training

Figure 4: Visualization of water exemplars before and after training. (a) Initial colors of the exemplar nodes, selected to cover a range of typical water body appearances. (b) Exemplar representations after training, where node colors are automatically adjusted to better reflect the actual water colors encountered, and edges represent graph connectivity. After training, the exemplars tend to be darker and more similar, reflecting the predominantly dimmer and closer hues of real underwater scenes compared to the diverse initial palette in (a).

Then, the normalized attention weight $\alpha_{ij}$ is obtained by applying softmax over neighbors of node $i$, i.e., $\mathcal{N}(i)$. The updated node feature is then computed as:

$$\tilde{\mathbf{w}}_i = \text{MLP}\left(\sum_{j \in \mathcal{N}(i)} \alpha_{ij} \cdot \mathbf{Z}_j\right) \in \mathbb{R}^C, \tag{13}$$

where $\text{MLP}(\cdot)$ is a shared projection module that maps hidden features to the token feature dimension $C$.

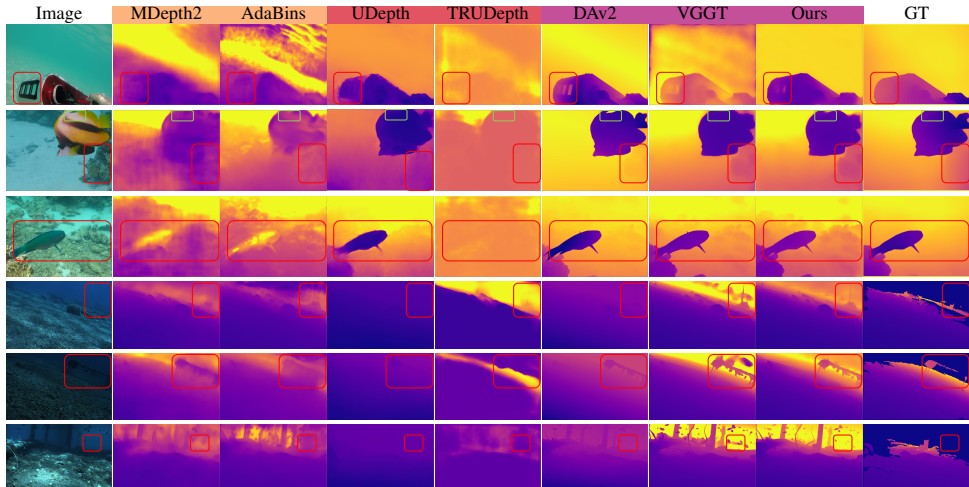

Figure 5: Comparison of depth predictions in terms of object-level shape accuracy (first 3 rows) and scene-level structural realism (last 3 rows). Regions with noticeable differences across predictions are highlighted with red bounding boxes.

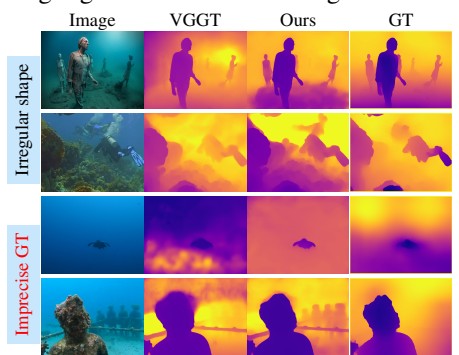

Figure 6: Foreground object shape comparison on USOD10K. Top: samples with irregular or complex object shapes. Bottom: samples with less accurate dataset-provided GTs, which may not fully reflect the actual scene structure.

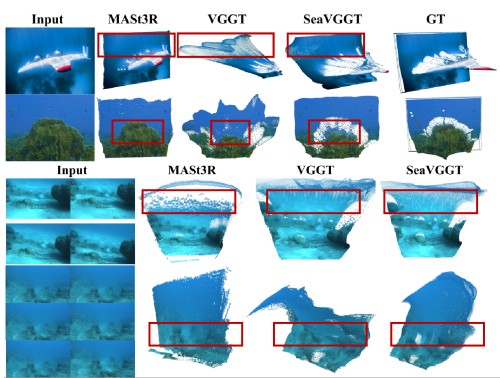

Figure 7: Qualitative point cloud comparison with VGGT and MASt3R. SeaVGGT exhibits more geometrically accurate 3D structures than VGGT and MASt3R.

### 3.4 PROTOTYPE-GUIDED TOKEN MODULATION

**Context-Aware Prototype Selection.** To respond to the current underwater condition, we first compute the similarity between the predicted global water color $\mathbf{B}$ and each prototype color $\mathbf{A}_i$:

$$s_i = \exp\left(-\gamma \|\mathbf{B} - \mathbf{A}_i\|_2\right), \tag{14}$$

where $\gamma$ is a temperature hyperparameter (empirically set to 10). We identify the top-$k$ most similar prototypes, indexed by $\mathcal{K}$, and normalize their scores via softmax: $\omega_i = \frac{s_i}{\sum_{j \in \mathcal{K}} s_j}$, $i \in \mathcal{K}$. The selected top-$k$ prototypes serve as *context-aware conditions*, providing a soft representation of the scene's water type. Each context-aware prototype $\tilde{\mathbf{w}}_i$ (refined by GAT) conditions a dedicated token modulation branch.

**Conditional Token Modulation.** Let input features be $\mathbf{X} = \{\mathbf{x}_k \in \mathbb{R}^C\}_{k=1}^K$, where $K$ is the number of tokens. Given a selected condition prototype $\tilde{\mathbf{w}}_i$, we apply hybrid modulation (see Figure 3) as follows:

*1) Global Modulation.* We interpolate between a global learnable prototype $\mathbf{w}_{\text{global}}$ and the selected prototype $\tilde{\mathbf{w}}_i$ to produce a global modulation vector:

$$\mathbf{g} = \sigma(\theta) \cdot \tilde{\mathbf{w}}_i + (1 - \sigma(\theta)) \cdot \mathbf{w}_{\text{global}}, \tag{15}$$

$$\theta = \frac{1}{C} \left\langle \mathbf{w}_{\text{global}}, \tilde{\mathbf{w}}_i \right\rangle \tag{16}$$

Table 1: Depth estimation metrics across different underwater scenes in the FLSea dataset. Recent works Yang et al. (2024a); Zhang et al. (2024b) are closely related for comparison, but since their depth estimation codes are not publicly available and they did not conduct experiments on FLSea, we are unable to report their results. The methods are grouped into three categories: terrestrial pretrained models fine-tuned with underwater data , underwater pretrained models , and foundation models performing zero-shot prediction . The complete quantitative results are provided in Table 10.

| Scene | Metric | MDepth2 | AdaBins | UDepth | TRUDepth | DAv2 | VGGT | Ours |
|---|---|---|---|---|---|---|---|---|
| Big Dice Loop | MAE↓ | 0.6715 | 0.7243 | 0.9177 | 1.0717 | 0.8519 | 0.4383 | **0.3623** |
| | RMSE↓ | 1.0719 | 1.1848 | 1.4395 | 1.4857 | 1.4292 | 0.8107 | **0.7703** |
| Coral Table Loop | MAE↓ | 0.6378 | 0.7560 | 0.6565 | 1.0159 | 0.6313 | 0.6602 | **0.5013** |
| | RMSE↓ | 0.8715 | 0.9710 | 0.8807 | 1.3396 | 0.8755 | 0.9459 | **0.7219** |
| Cross Pyramid Loop | MAE↓ | 0.4705 | 0.6007 | 0.5585 | 0.8491 | 0.4927 | 0.4040 | **0.2607** |
| | RMSE↓ | 0.6121 | 0.7537 | 0.7269 | 1.0809 | 0.6669 | 0.5157 | **0.3768** |
| Dice Path | MAE↓ | 0.4738 | 0.5721 | 0.7376 | 0.8140 | 0.6464 | 0.3742 | **0.3162** |
| | RMSE↓ | 0.6150 | 0.7240 | 0.9663 | 1.0212 | 0.8930 | 0.5505 | **0.4906** |
| Northeast Path | MAE↓ | 0.6804 | 0.8736 | 1.1590 | 1.2501 | 1.1241 | 0.6681 | **0.5877** |
| | RMSE↓ | 1.0253 | 1.1667 | 1.5714 | 1.6532 | 1.6157 | 0.9699 | **0.9197** |
| Pier Path | MAE↓ | 0.6384 | 0.5493 | 0.7328 | 0.9908 | 0.6553 | 0.3531 | **0.2823** |
| | RMSE↓ | 0.8656 | 0.7494 | 1.0166 | 1.2629 | 0.9421 | 0.5359 | **0.4692** |
| Sub Pier | MAE↓ | 0.5370 | 0.5440 | 0.6679 | 0.8937 | 0.6295 | 0.4174 | **0.3538** |
| | RMSE↓ | 0.8209 | 0.7825 | 0.9887 | 1.2396 | 0.9747 | 0.6503 | **0.5603** |

Table 2: Depth estimation results on USOD10K and SQUID. ↑: higher is better, ↓: lower is better.

| Metric | USOD10K (mono) | | | | | | | SQUID (stereo) | |
|---|---|---|---|---|---|---|---|---|---|
| | MDepth2 | AdaBins | UDepth | TRUDepth | DAv2 | VGGT | Ours | VGGT | Ours |
| MAE ↓ | 1.7220 | 1.8727 | 2.0539 | 1.7055 | 2.1052 | 1.5248 | **1.3414** | 0.1213 | **0.0968** |
| RMSE ↓ | 2.0938 | 2.2826 | 2.4070 | 2.0349 | 2.4789 | 1.8534 | **1.6595** | 0.1774 | **0.1403** |
| REL ↓ | 1.0300 | 1.1484 | 0.5464 | 1.4269 | 0.7031 | 1.2582 | **1.1415** | 0.2452 | **0.1949** |
| $\delta_1$ ↑ | 0.3450 | 0.3361 | 0.2616 | 0.3634 | 0.3088 | 0.4319 | **0.4835** | 0.5238 | **0.7013** |
| $\delta_2$ ↑ | 0.5933 | 0.5742 | 0.5141 | 0.6436 | 0.5690 | 0.6792 | **0.7249** | 0.8407 | **0.9102** |
| $\delta_3$ ↑ | 0.7423 | 0.7190 | 0.7200 | 0.7957 | 0.7526 | 0.8021 | **0.8272** | **0.9654** | 0.9651 |
| si-RMSE ↓ | 0.6842 | 0.7236 | **0.4348** | 0.7185 | 0.5107 | 0.6619 | 0.6253 | 0.2232 | **0.1935** |

Each token is then modulated via channel-wise multiplication: $\mathbf{x}_k \leftarrow \mathbf{x}_k \odot \mathbf{g}$.

*2) Local Modulation.* A local prototype matrix $\mathbf{W}_{local} \in \mathbb{R}^{K \times C}$ is used to capture token-level variation. For each token, a query vector $\mathbf{q}_k$ is extracted and used to compute local modulation via attention:

$$\mathbf{b}_k = \sum_{i=1}^{K} \text{softmax}\left(\frac{\mathbf{q}_k^\top \mathbf{w}_{local,i}}{\sqrt{C}}\right) \cdot \mathbf{w}_{local,i} \qquad (17)$$

*3) Spatial Fusion.* A two-layer CNN generates per-token weights $\alpha_k \in [0, 1]$ to fuse global and local outputs:

$$\tilde{\mathbf{x}}_k = \alpha_k \cdot (\mathbf{x}_k \odot \mathbf{g}) + (1 - \alpha_k) \cdot \mathbf{b}_k \qquad (18)$$

The modulated tokens $\tilde{\mathbf{X}} = \{\tilde{\mathbf{x}}_k\}_{k=1}^{K}$ are passed to the decoder for prediction.

**Prototype-Conditioned Physical Prediction.** Each condition prototype also predicts a sample-specific physical parameter $\boldsymbol{\beta}_i$ via an MLP. The final estimate aggregates top-$K$ predictions by similarity-weighted averaging: $\boldsymbol{\beta}_{\text{final}} = \sum_{i \in \mathcal{K}} \frac{s_i}{\sum_{j \in \mathcal{K}} s_j} \cdot \boldsymbol{\beta}_i$.

**Discussion on Modulation and Prototype Graph.** Our water-prototype guided modulation operates hierarchically, combining **global water-type-aware scaling** with **local fine-grained adjustment**. Global modulation adapts feature representations to overall environmental conditions, while local modulation ensures spatially precise adaptation based on local content. This design

Table 3: Inference time (Single A6000 GPU) and parameter count.

| Method | Time (ms) | Params |
|--------|-----------|--------|
| VGGT | 2.06 | 1.26B |
| Ours | 2.39 | 1.34B |

Table 4: Ablation on loss functions.

| $\mathcal{L}_{rec}$ | $\mathcal{L}_{enh}$ | $\mathcal{L}_{color}$ | $\mathcal{L}_{bg}$ / $\mathbb{E}[\mathbf{I}]$ | MAE↓ | RMSE↓ |
|---|---|---|---|---|---|
| ✓ | | | $\mathcal{L}_{bg}$ | 1.6484 | 2.0471 |
| ✓ | ✓ | | $\mathcal{L}_{bg}$ | 1.5241 | 1.8335 |
| ✓ | ✓ | ✓ | $\mathcal{L}_{bg}$ | 1.3414 | 1.6595 |
| ✓ | ✓ | ✓ | $\mathbb{E}[\mathbf{I}]$ | 1.5776 | 1.9114 |
| ✓ | ✓ | ✓ | | 1.7442 | 2.4329 |

Table 5: Analysis of depth supervision under noisy conditions.

| Metric | 5% | 10% | 20% | 30% |
|--------|------|------|------|------|
| MAE | 1.3753 | 1.3740 | 1.4293 | 1.8256 |
| RMSE | 1.6781 | 1.7108 | 1.7571 | 2.6844 |

Table 6: Ablation on initialization strategies.

| Initialization | Training Steps | MAE | RMSE |
|---|---|---|---|
| Prior | 20000 | 1.3414 | **1.6595** |
| Random #1 | 20000 | 1.4879 | 1.8012 |
| Random #1 | 50000 | 1.3660 | 1.7011 |
| Random #2 | 50000 | **1.3353** | 1.6818 |

Table 7: Ablation on color distance thresholds.

| $\tau$ | MAE | RMSE |
|---|---|---|
| 0.1 | 1.3641 | 1.6986 |
| 0.3 | 1.3414 | 1.6595 |
| 0.6 | 1.3577 | 1.8915 |
| 1.0 | 1.4823 | 2.1089 |

Table 8: Ablation on prototype interaction strategies.

| Interaction | #Layers | MAE | RMSE |
|---|---|---|---|
| Graph Attention | 1 | 1.3414 | 1.6595 |
| Nearest Neighbor | 0 | 1.4602 | 1.8057 |
| Triplet Loss + Weighted Sum | 0 | 1.5027 | 1.8734 |

effectively bridges high-level priors and low-level details, supporting robust and generalizable geometric estimation across diverse underwater scenes, and is applied *without retraining the VGGT backbone or prediction heads*, enabling efficient adaptation.

The effectiveness of this modulation is reinforced by the learned **prototype graph** (Figure 4). Visualization of the prototypes and their adjacency matrix shows structured and meaningful connections, indicating that the model organizes representative water appearance patterns through prototype interactions. This structured graph supports the hierarchical modulation by providing interpretable priors that guide both global and local feature adaptation, thereby enhancing depth prediction quality and maintaining physical consistency in challenging underwater environments.

## 4 EXPERIMENTS

**Datasets.** We evaluate SeaVGGT on three underwater datasets: FLSea Randall (2023) (22,451 RGB images with metric depth in shallow water, featuring light attenuation, turbidity, and specular reflections), USOD10K Hong et al. (2023) (10,255 images of 70 salient object classes across 12 scenes; evaluation on 3,077 test images without using training data), and SQUID Berman et al. (2021) (57 stereo pairs from four sites in Israel, covering coral reefs, a shipwreck, and rocky reefs at 3–30m depth).

**Experimental Configurations.** We adopt a lightweight training strategy with only three sets of trainable parameters: (1) 24 learnable water prototypes along with their corresponding modulation parameters, (2) a lightweight CNN designed for background light estimation, and (3) an image enhancement head (DPT head Ranftl et al. (2021)), which is used solely during training and removed at inference time. All models are optimized using the Adam optimizer with a learning rate of $1 \times 10^{-4}$ and trained on a single NVIDIA A6000 GPU with a batch size of 1. In the prototype interaction module, the top-$k$ selection is set to $k = 4$. We evaluate using standard metrics commonly adopted in depth estimation, and perform scale alignment between the predicted depth and the ground-truth depth for each method. The training set consists of 11,400 unlabeled underwater images from the MVK Dataset Du et al. (2024). To enhance generalization, we apply standard data augmentation techniques, including random horizontal and vertical flipping, as well as color jittering. The overall training loss is formulated as a weighted sum of four components: the reconstruction loss and the enhancement loss (both with weight 1.0), the color consistency loss (weight 0.05), and the background prediction loss (weight 0.1). Among the enhancement loss components, the weights for the L1 loss, SSIM loss, perceptual loss, and gradient loss are set to 1, 0.1, 0.1, and 0.1, respectively.

### 4.1 MAIN RESULTS

In the underwater depth prediction experiments, we conducted comprehensive comparisons with a range of existing methods grouped into three categories: 1) terrestrial pretrained models fine-tuned on FLSea underwater images and depth labels (MDepth2 Godard et al. (2019), AdaBins Bhat et al.

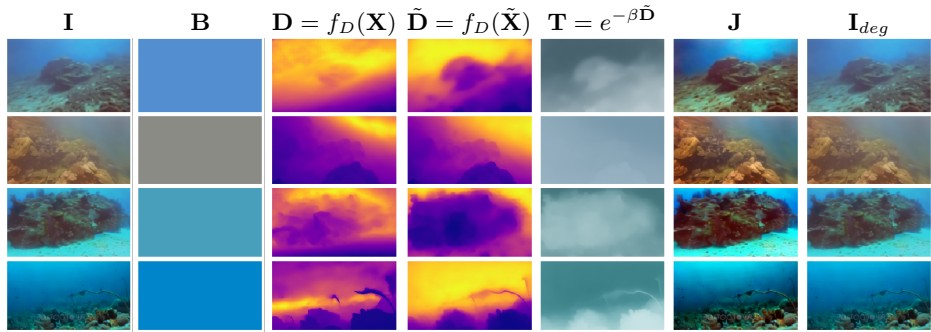

Figure 8: Visualization of intermediate variables in the water-prototype guided modulation pipeline.

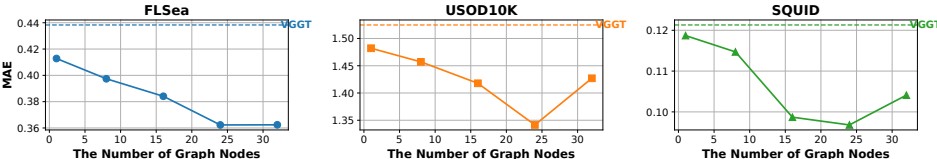

Figure 9: Ablation study on the number of graph nodes.

(2021)); 2) underwater pretrained models specifically trained on underwater depth data (UDepth Yu et al. (2022), TRUDepth Ebner et al. (2024)); and 3) foundation models evaluated in a zero-shot manner on underwater scenes without any underwater depth annotations (DAv2 Yang et al. (2024b), VGGT). Our method belongs to this last category, leveraging foundation models for zero-shot depth prediction without relying on depth supervision. This grouping emphasizes the differing levels of prior knowledge and supervision among the compared approaches and positions our method in the context of generalizable, annotation-free underwater depth estimation.

Table 1 presents the results of monocular depth estimation methods evaluated on multiple scenes from the FLSea dataset. Our method achieves the best performance across all metrics, significantly outperforming VGGT by a substantial margin.

Table 2 demonstrates that our method achieves state-of-the-art performance in underwater depth estimation on the challenging USOD10K dataset, outperforming VGGT across multiple evaluation metrics. To further validate the effectiveness of our approach, we present qualitative 3D visualizations comparing our predicted point clouds with those from competing methods (e.g., MASt3R Leroy et al. (2024)) in Figure 7. These visualizations clearly highlight our method's superior capability to recover accurate and coherent 3D scene structures in complex underwater environments. Moreover, Table 2 shows that our approach achieves significant improvements over VGGT on stereo settings in the SQUID dataset, despite not being trained under multi-view conditions, thereby demonstrating the generality and effectiveness of our token modulation strategy.

**Computational Complexity.** Table 3 shows that our full model incorporates two lightweight modules on top of VGGT: the modulation part with 75.8 million parameters and a compact background light CNN containing only 33K parameters, resulting in a total of approximately 1.34 billion parameters. Despite this increase, the inference time remains efficient, with only a modest rise from 2.06 ms for VGGT to 2.39 ms for our model.

**Visualization and Analysis of Intermediate Variables.** Figure 8 visualizes key intermediate outputs. The initial VGGT depth $\mathbf{D}$ is noisy, whereas the modulated depth $\tilde{\mathbf{D}}$ shows clearer boundaries and stronger geometric consistency. The transmission map $\mathbf{T}$ models depth-dependent attenuation, enabling realistic degraded images $\mathbf{I}_{\text{deg}}$, and the reconstructed clean image $\mathbf{J}$ exhibits improved contrast. These results illustrate our self-supervised feedback mechanism: modulation adjusts tokens $\tilde{\mathbf{X}}$ so that the frozen depth head produces $\tilde{\mathbf{D}}$ consistent with $\mathbf{J}$. Reconstruction losses are minimized only when the two align, indicating that the modulation is effective, physics-guided, and improves depth quality through cross-modal consistency.

## 4.2 ABLATION STUDY

**Effect of the Number of Graph Nodes.** We conduct an ablation study to investigate how the number of graph nodes (i.e., water prototypes) affects geometric estimation. As shown in

Figure 9, increasing the number of graph nodes improves performance up to a certain point, with diminishing returns beyond 24 nodes. Notably, our method consistently outperforms VGGT across all configurations, demonstrating the robustness and generalization benefit of our water-prototype modulation.

**Effect of the Loss Design and Weight.** We conduct an ablation study to assess the contribution of each loss component (Table 4). Using only the reconstruction loss $\mathcal{L}$rec yields higher errors, while adding the enhancement loss $\mathcal{L}$enh noticeably improves results. Incorporating the color consistency loss $\mathcal{L}$color provides further gains by enforcing realistic underwater color priors. Replacing the naive background estimate $\mathbb{E}[\mathbf{I}]$ with our background loss $\mathcal{L}$bg reduces error, confirming its effectiveness. We additionally varied the weight of each loss term while keeping others fixed. As shown in Figure 10, performance remains stable across a broad range of weight values, demonstrating that our improvements are robust rather than the outcome of a specific hyperparameter setting.

**Sensitivity to Noise from Imperfect Depth Maps.** We further evaluate robustness by injecting proportional noise into $\mathbf{D}$ during training (Table 5). While very large noise (e.g., $\sigma = 30\%$) slightly degrades $\mathbf{B}$ estimation and final depth accuracy, performance remains stable under realistic noise levels ($\sigma \leq 20\%$), indicating that the proxy depth supervision is reasonably robust.

**Effect of Prototype Graph Initialization.** To assess sensitivity to initialization, we conducted an ablation study comparing the prior-based initialization with two independent random initializations, denoted as Random #1 and Random #2. The results are shown in Table 6, indicating that random initialization can lead to slower convergence. However, after sufficient training steps, the final performance recovers to a level comparable with prior-based initialization, demonstrating the model's robustness to the choice of initial prototype colors. Furthermore, we visualized the learned prototype graphs and adjacency matrices for the two random initializations (see Figure 14). Despite starting from random colors,

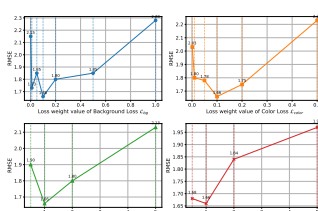

Figure 10: Sensitivity of Depth Prediction to Loss Weight (USOD10K).

the converged prototype graph still exhibits clear structural regularities, and the adjacency matrix reveals consistent connectivity patterns among graph nodes.

**Sensitivity to Color Distance Threshold.** To evaluate sensitivity to this hyperparameter, we varied $\tau$ and report the results in Table 7. The model performs best at $\tau = 0.3$, which is used in our main experiments. Smaller values slightly weaken graph connectivity, while larger values may introduce less relevant edges; however, the impact in both cases is limited because edge weights mitigate noise. Overall, the model shows low sensitivity to the choice of $\tau$.

**Effect of the GNN.** To evaluate the ability to model higher-order interactions among prototypes, we implemented a metric-learning based baseline that operates at the image level. Since we do not have ground-truth prototype assignments, we follow unsupervised metric learning practices and treat the nearest prototype as a pseudo-positive, and the others as pseudo-negatives: $\mathcal{L}_{\mathrm{metric}} = \max(0, d(z, p_+) - d(z, p_-) + m)$, where $p_+$ is the closest prototype for the current image and $p_-$ is sampled from the remaining prototypes. After training, scattering parameters are estimated as a convex combination $\sum_k \alpha_k \, p_k = \sum_k \frac{\exp(-d(z, p_k))}{\sum_j \exp(-d(z, p_j))} \, p_k$. Table 7 compares several prototype interaction strategies, supporting our design rationale: GNNs capture structured inter-prototype relationships and propagate information, leading to more stable and accurate depth predictions. In contrast, metric learning ignores these relationships, and nearest-neighbor methods cannot generalize to unseen water types.

## 5 CONCLUSION

This paper introduces SeaVGGT, a lightweight and effective adaptation of the VGGT framework for underwater scenarios. Without relying on labeled data or modifying the VGGT backbone and prediction heads, SeaVGGT leverages a self-supervised strategy to achieve robust domain adaptation. Our method yields clear improvements in 3D structure recovery, object boundary delineation, and multi-view consistency, all while maintaining low computational cost. These advantages make SeaVGGT a practical solution for a wide range of real-world underwater applications.

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

# A APPENDIX

## A.1 EFFECT OF THE LOSS WEIGHT.

To investigate the impact of different loss weights on depth prediction, we performed an ablation study by varying the weights of four loss components: Background Loss ($\mathcal{L}bg$), Color Loss ($\mathcal{L}color$), Reconstruction Loss ($\mathcal{L}rec$), and Enhancement Loss ($\mathcal{L}enh$). The evaluation metric is RMSE on the USOD10K dataset.

The results in Figure 10, Figure 11, and Table 9 show that the model performs consistently well across a wide range of loss weights. While moderate weights generally yield slightly better RMSE, the overall performance is not highly sensitive to the exact values of each loss coefficient. This indicates that our method is robust and does not require careful tuning of individual loss weights to achieve stable depth estimation.

Table 9: RMSE values under different loss weights for USOD10K

| Background Loss | | Color Loss | | Reconstruction Loss | | Enhancement Loss | |
|---|---|---|---|---|---|---|---|
| $\lambda_{bg}$ | RMSE | $\lambda_{color}$ | RMSE | $\lambda_{rec}$ | RMSE | $\lambda_{enh}$ | RMSE |
| 0.001 | 2.15 | 0.001 | 2.03 | 0.5 | 1.90 | 0.5 | 1.68 |
| 0.01 | 1.73 | 0.01 | 1.80 | 1.0 | 1.66 | 1.0 | 1.66 |
| 0.05 | 1.85 | 0.05 | 1.78 | 2.0 | 1.80 | 2.0 | 1.84 |
| 0.1 | 1.66 | 0.1 | 1.66 | 5.0 | 2.13 | 5.0 | 1.97 |
| 0.2 | 1.80 | 0.2 | 1.75 | | | | |
| 0.5 | 1.85 | 0.5 | 2.23 | | | | |
| 1.0 | 2.28 | | | | | | |

### A.2 MORE VISUALIZATION ANALYSIS OF VARIABLES

To further elucidate the effectiveness of our proposed modulation mechanism and its alignment with underwater image formation principles, we provide a detailed analysis of key intermediate variables visualized along the processing pipeline in Figure 12. These variables include the initial ($\mathbf{D}$) and refined depth maps $\tilde{\mathbf{D}}$, transmission map $\mathbf{T}$, degraded image synthesis $\mathbf{I}_{deg}$, and reconstructed clean image $\mathbf{J}$. The relationships among these variables are grounded in the physical model of underwater imaging and play a critical role in enabling self-supervised learning. Below, we explain each component in detail and describe how they interact.

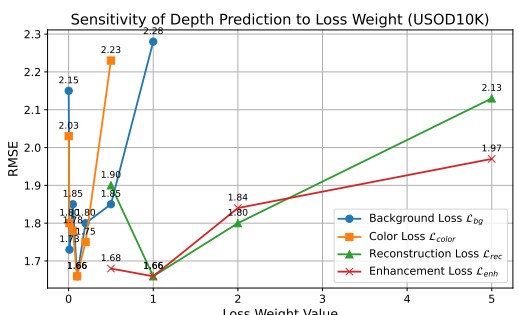

Figure 11: Sensitivity of Depth Prediction to Loss Weight (USOD10K).

**Initial Depth Map $\mathbf{D} = f_D(\mathbf{X})$.** The variable $\mathbf{D}$ represents the initial dense depth prediction produced by the pretrained foundational model VGGT Wang et al. (2025). This depth is inferred directly from the tokens $\mathbf{X}$ of the input underwater image $\mathbf{I}$, which is typically affected by scattering, wavelength-dependent attenuation, and reduced visibility.

**Refined Depth Map $\tilde{\mathbf{D}} = f_D(\tilde{\mathbf{X}})$.** The variable $\tilde{\mathbf{D}}$ denotes the refined depth output after applying the proposed prototype-guided token modulation mechanism. This step is context-aware and leverages a water-prototype graph to adaptively condition token features based on water color and scattering characteristics, resulting in improved depth predictions under different underwater conditions.

**Transmission Map $\mathbf{T}$.** The transmission map $\mathbf{T}$ is computed from the refined depth $\tilde{\mathbf{D}}$ using an underwater attenuation model:

$$\mathbf{T} = e^{-\beta\tilde{\mathbf{D}}}, \tag{19}$$

where $\beta$ is an attenuation coefficient specific to the wavelength (or water type), and $x$ indexes the pixel location. The transmission map models how much light from the scene reaches the camera after traveling through the water column, and it is depth-dependent.

**Reconstructed Clean Image $\mathbf{J}$.** The clean image $\mathbf{J}$ is reconstructed by inverting the underwater image formation model, using the estimated refined depth map $\tilde{\mathbf{D}}$ and background color $\mathbf{B}$:

$$\mathbf{J} = \frac{\mathbf{I} - \mathbf{B} \odot (1 - \mathbf{T})}{\mathbf{T} + \epsilon}, \tag{20}$$

where $\mathbf{T} = e^{-\beta\tilde{\mathbf{D}}}$ is the transmission map derived from depth, and $\epsilon$ is a small constant to avoid division by zero. The reconstructed $\mathbf{J}$ represents an estimate of the visibility-restored scene, compensating for both depth-dependent attenuation and color cast caused by scattering.

Table 10: Depth estimation metrics on different underwater scenes in FLSea.

| Scene | Metric | MDepth2 | AdaBins | UDepth | TRUDepth | DAv2 | VGGT | Ours |
|-------|--------|---------|---------|--------|----------|------|------|------|
| Big Dice Loop | MAE↓ | 0.6715 | 0.7243 | 0.9177 | 1.0717 | 0.8519 | 0.4383 | **0.3623** |
| | RMSE↓ | 1.0719 | 1.1848 | 1.4395 | 1.4857 | 1.4292 | 0.8107 | **0.7703** |
| | REL↓ | 0.2132 | 0.2409 | 0.2895 | 0.3073 | 0.2442 | 0.1295 | **0.0880** |
| | si-RMSE↓ | 0.1750 | 0.1885 | 0.5546 | 0.3583 | 0.3723 | 0.1077 | **0.0997** |
| Coral Table Loop | MAE↓ | 0.6378 | 0.7560 | 0.6565 | 1.0159 | 0.6313 | 0.6602 | **0.5013** |
| | RMSE↓ | 0.8715 | 0.9710 | 0.8807 | 1.3396 | 0.8755 | 0.9459 | **0.7219** |
| | REL↓ | 0.2214 | 0.2775 | 0.2423 | 0.3691 | 0.2110 | 0.2107 | **0.1592** |
| | si-RMSE↓ | 0.2075 | 0.2446 | 0.4094 | 0.3931 | 0.2856 | 0.2078 | **0.1522** |
| Cross Pyramid Loop | MAE↓ | 0.4705 | 0.6007 | 0.5585 | 0.8491 | 0.4927 | 0.4040 | **0.2607** |
| | RMSE↓ | 0.6121 | 0.7537 | 0.7269 | 1.0809 | 0.6669 | 0.5157 | **0.3768** |
| | REL↓ | 0.2067 | 0.2882 | 0.2624 | 0.3932 | 0.2135 | 0.1789 | **0.1028** |
| | si-RMSE↓ | 0.1905 | 0.2508 | 0.4524 | 0.4062 | 0.3073 | 0.1592 | **0.1056** |
| Dice Path | MAE↓ | 0.4738 | 0.5721 | 0.7376 | 0.8140 | 0.6464 | 0.3742 | **0.3162** |
| | RMSE↓ | 0.6150 | 0.7240 | 0.9663 | 1.0212 | 0.8930 | 0.5505 | **0.4906** |
| | REL↓ | 0.1874 | 0.2262 | 0.2899 | 0.3156 | 0.2348 | 0.1415 | **0.1121** |
| | si-RMSE↓ | 0.1740 | 0.1870 | 0.5056 | 0.3278 | 0.3776 | 0.1170 | **0.1106** |
| Northeast Path | MAE↓ | 0.6804 | 0.8736 | 1.1590 | 1.2501 | 1.1241 | 0.6681 | **0.5877** |
| | RMSE↓ | 1.0253 | 1.1667 | 1.5714 | 1.6532 | 1.6157 | 0.9699 | **0.9197** |
| | REL↓ | 0.1629 | 0.2632 | 0.3131 | 0.3323 | 0.2824 | 0.1900 | **0.1433** |
| | si-RMSE↓ | 0.1509 | 0.1973 | 0.5216 | 0.3518 | 0.3601 | 0.1578 | **0.1416** |
| Pier Path | MAE↓ | 0.6384 | 0.5493 | 0.7328 | 0.9908 | 0.6553 | 0.3531 | **0.2823** |
| | RMSE↓ | 0.8656 | 0.7494 | 1.0166 | 1.2629 | 0.9421 | 0.5359 | **0.4692** |
| | REL↓ | 0.2227 | 0.2003 | 0.2521 | 0.3532 | 0.2133 | 0.1113 | **0.0868** |
| | si-RMSE↓ | 0.2185 | 0.1788 | 0.4564 | 0.3604 | 0.3632 | 0.1056 | **0.0957** |
| Sub Pier | MAE↓ | 0.5370 | 0.5440 | 0.6679 | 0.8937 | 0.6295 | 0.4174 | **0.3538** |
| | RMSE↓ | 0.8209 | 0.7825 | 0.9887 | 1.2396 | 0.9747 | 0.6503 | **0.5603** |
| | REL↓ | 0.2382 | 0.2727 | 0.2999 | 0.4275 | 0.2650 | 0.1928 | **0.1474** |
| | si-RMSE↓ | 0.2145 | 0.2127 | 0.4823 | 0.3853 | 0.3641 | 0.1594 | **0.1325** |

Importantly, this reconstruction process is only valid under the assumption that the estimated depth $\tilde{D}$ and background color $B$ are accurate. If either component is unreliable, the resulting $J$ will contain artifacts or unrealistic content. Conversely, when $\tilde{D}$ captures the true scene geometry and $B$ correctly models ambient water color, the forward synthesis of the degraded image $I_{deg}$, computed from $J$ and $T$, will closely match the original observation $I$. This consistency underpins the self-supervised learning objective and reinforces physically grounded depth estimation.

## A.3 FURTHER RESULTS AND COMPARISONS

**Monocular Point Cloud Comparison**  Figure 15 presents qualitative comparisons of 3D point clouds reconstructed from monocular images, using VGGT, MASt3R, and our method. This visualization offers an intuitive perspective on the geometric structure captured by each method. As shown, both VGGT Wang et al. (2025) and MASt3R Leroy et al. (2024) produce noisy and spatially fragmented point clouds when applied directly to underwater images, reflecting their difficulty in generalizing across severe underwater degradations. These reconstructions often exhibit distorted object shapes, surface discontinuities, and depth inconsistency, particularly in regions affected by scattering or color cast.

In contrast, our method yields significantly more coherent and physically plausible 3D structures. The resulting point clouds exhibit sharper object boundaries, smoother surfaces, and more accurate spatial layouts. This improvement highlights the effectiveness of our water-aware modulation mechanism and self-supervised learning framework in promoting geometrically consistent depth estimation, even without access to ground-truth depth or labels. Such improvements are critical for downstream tasks like underwater mapping, navigation, or manipulation in real-world marine environments.

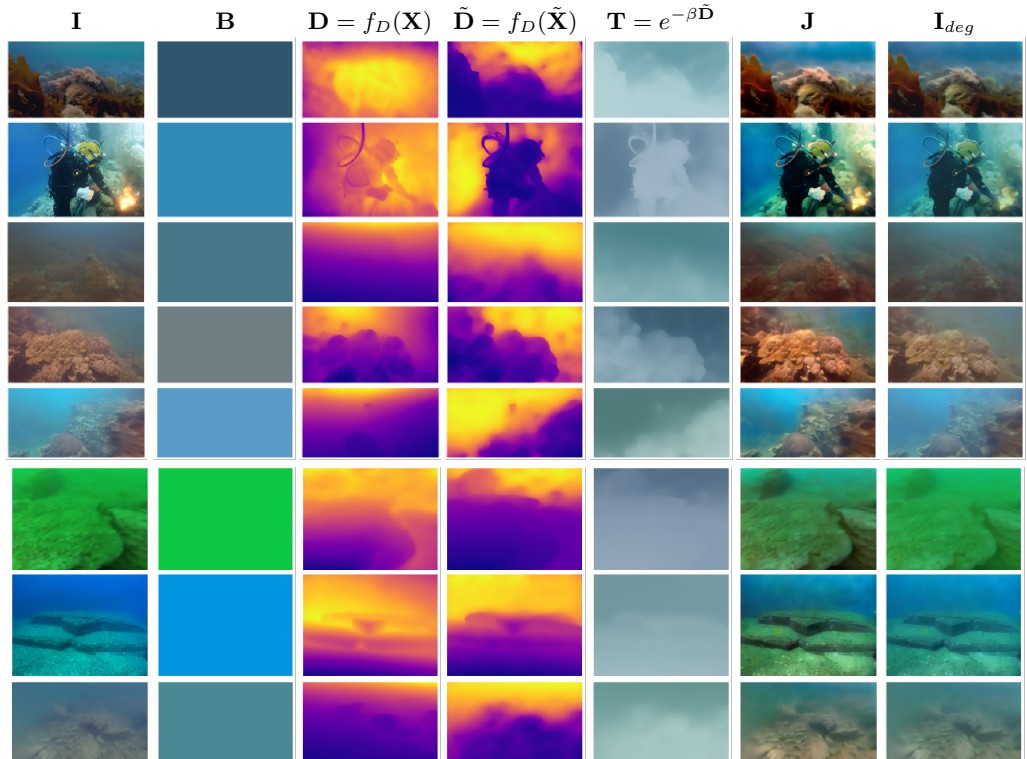

Figure 12: Visualization of intermediate variables in the water-prototype guided modulation pipeline.

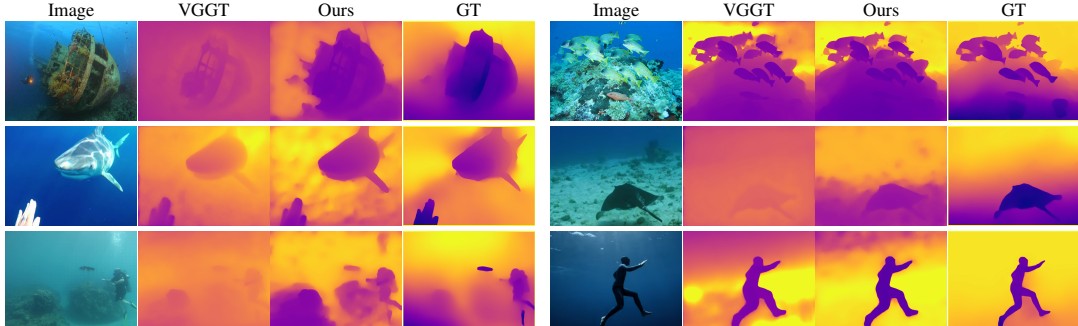

Figure 13: Foreground object shape comparison on USOD10K. Our method produces clearer object shapes and more realistic background depth.

**Stereo Point Cloud Comparison.** Figure 16 illustrates qualitative comparisons of 3D point clouds reconstructed from stereo depth predictions using VGGT, MASt3R, and our proposed method. In this setting, stereo pairs are provided as input, allowing models to leverage binocular cues in addition to monocular priors.

While VGGT and MASt3R benefit from stereo input, their reconstructions still suffer from notable geometric artifacts in underwater scenes. Specifically, their point clouds tend to exhibit depth noise, surface distortions, and poor alignment with object boundaries—especially in regions affected by turbidity, color shift, or low texture. This reflects their limited robustness to underwater degradations even in the presence of stereo cues.

In contrast, our method consistently produces sharper and more structurally faithful point clouds. Surfaces are smoother, object boundaries are more distinct, and the overall spatial layout better preserves real-world scene geometry. This suggests that our water-aware modulation and unsupervised learning framework not only adapts well in monocular settings, but also complements

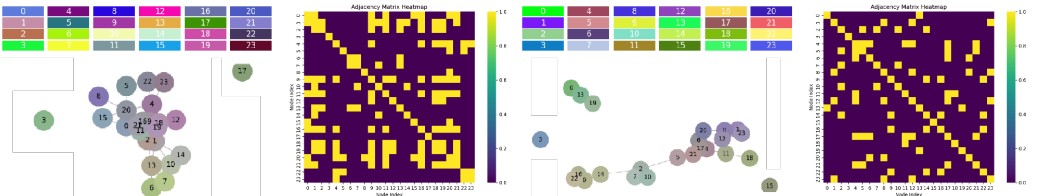

Figure 14: Effect of randomly initialized colors for the water exemplars. Despite the random initialization, the prototype graph learned after model convergence still exhibits clear structural regularities, and the adjacency matrix reveals the connectivity patterns among graph nodes.

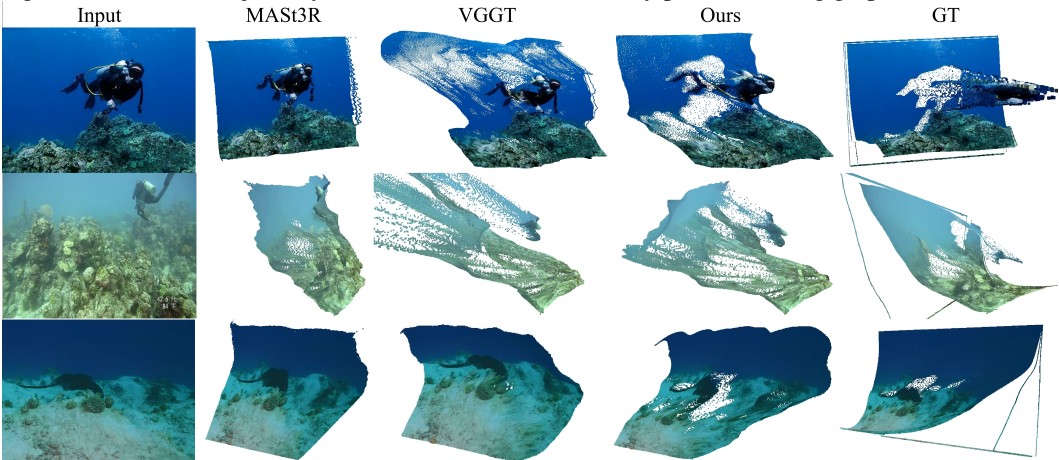

Figure 15: Qualitative monocular point cloud comparison with VGGT and MASt3R.

stereo-based depth estimation by enhancing feature reliability and enforcing better geometric consistency.

**Multi-view Point Cloud Comparison.** Figure 17 presents a qualitative comparison of multi-view reconstructed point clouds using our method versus VGGT.

This multi-view scenario emphasizes the importance of depth consistency across frames. As shown, the point clouds produced by VGGT often exhibit misalignments and accumulated clutter, resulting in fragmented or overlapping structures when multiple views are fused. These inconsistencies arise from per-frame prediction noise and are particularly pronounced in scattering-dominated underwater environments.

In contrast, our method produces cleaner and more structurally consistent point clouds under multi-view fusion. The alignment across frames is visibly more stable, with fewer redundant or misregistered points. This demonstrates that our water-prototype modulation contributes to improved temporal coherence, making the resulting 3D reconstruction more reliable for downstream applications such as underwater mapping, SLAM, or structure-from-motion.

**Multi-view 2D Point Tracking.** Figure 18 visualizes multi-view 2D point tracking results by reprojecting 3D points across frames using the predicted depths and known camera poses. This visualization serves as an indirect measure of inter-frame geometric consistency.

While tracking in underwater scenes is inherently challenging due to scattering, low texture, and depth ambiguity, our method shows more stable tracking trajectories compared to VGGT in several regions. In particular, the tracked points exhibit smoother motion across views and fewer abrupt jumps, suggesting improved depth continuity and cross-frame consistency.

Although some residual drift remains, especially in regions with limited visual cues, these results indicate that the proposed water-prototype modulation contributes to more coherent geometry over time, which may benefit applications such as structure-from-motion or feature-level association under water.

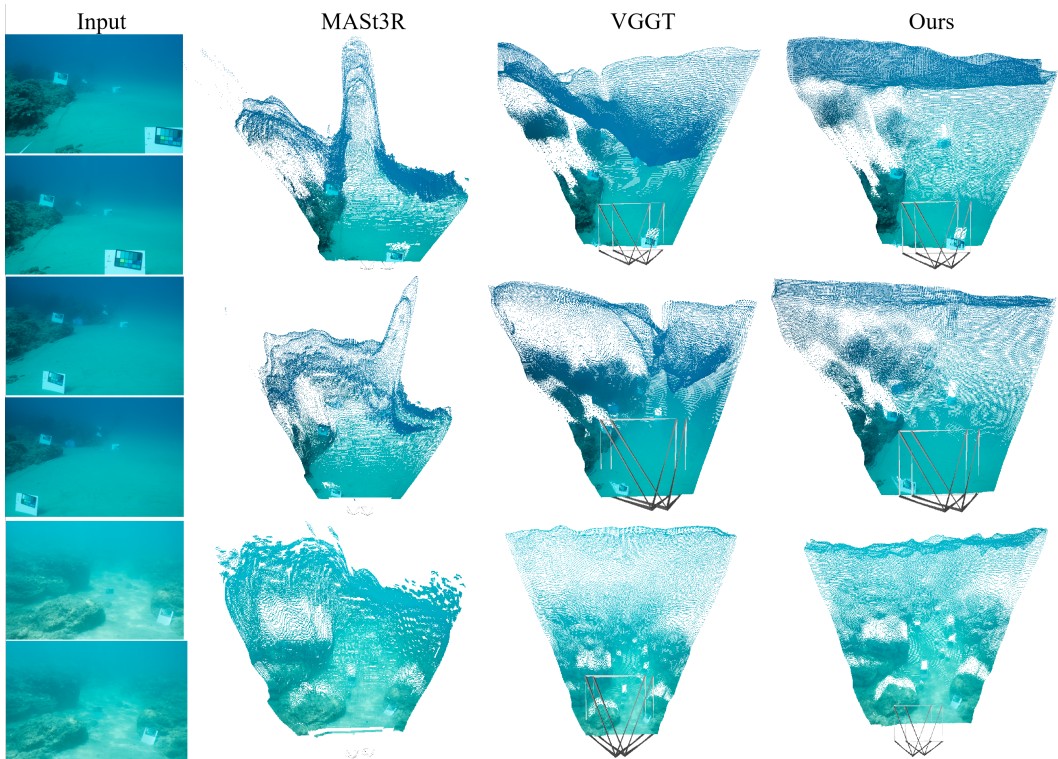

Figure 16: Qualitative stereo point cloud comparison with VGGT and MASt3R.

## B  ETHICS STATEMENT

This work adheres to the ICLR Code of Ethics. In this study, no human subjects or animal experimentation was involved. All datasets used, including our constructed Motion-labed dataset, were sourced in compliance with relevant usage guidelines, ensuring no violation of privacy. We have taken care to avoid any biases or discriminatory outcomes in our research process. No personally identifiable information was used, and no experiments were conducted that could raise privacy or security concerns. We are committed to maintaining transparency and integrity throughout the research process.

## C  REPRODUCIBILITY STATEMENT

We have made every effort to ensure that the results presented in this paper are reproducible. The core codes and datasets have been made publicly available in the Supplementary Material to facilitate replication and verification. The experimental setup, including training steps, model configurations, and hardware details, is described in detail in the paper. We have also provided a full description of our framework to assist others in reproducing our experiments.

Additionally, datasets used in the paper are publicly available, ensuring consistent and reproducible evaluation results.

We believe these measures will enable other researchers to reproduce our work and further advance the field.

## D  LLM USAGE

Large Language Models (LLMs) were used to aid in the writing and polishing of the manuscript. Specifically, we used an LLM to assist in refining the language, improving readability, and ensuring

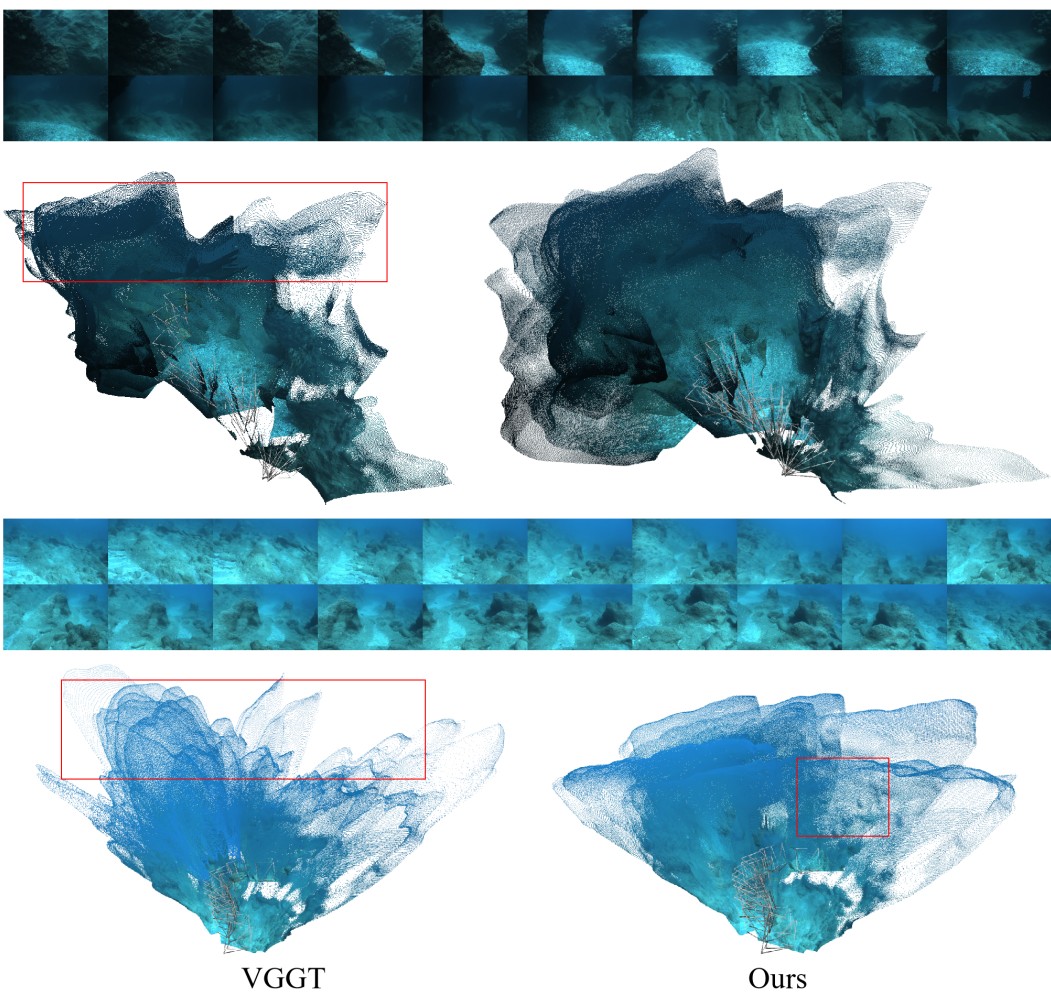

Figure 17: Qualitative multi-view point cloud comparison with VGGT.

clarity in various sections of the paper. The model helped with tasks such as sentence rephrasing, grammar checking, and enhancing the overall flow of the text.

It is important to note that the LLM was not involved in the ideation, research methodology, or experimental design. All research concepts, ideas, and analyses were developed and conducted by the authors. The contributions of the LLM were solely focused on improving the linguistic quality of the paper, with no involvement in the scientific content or data analysis.

The authors take full responsibility for the content of the manuscript, including any text generated or polished by the LLM. We have ensured that the LLM-generated text adheres to ethical guidelines and does not contribute to plagiarism or scientific misconduct.

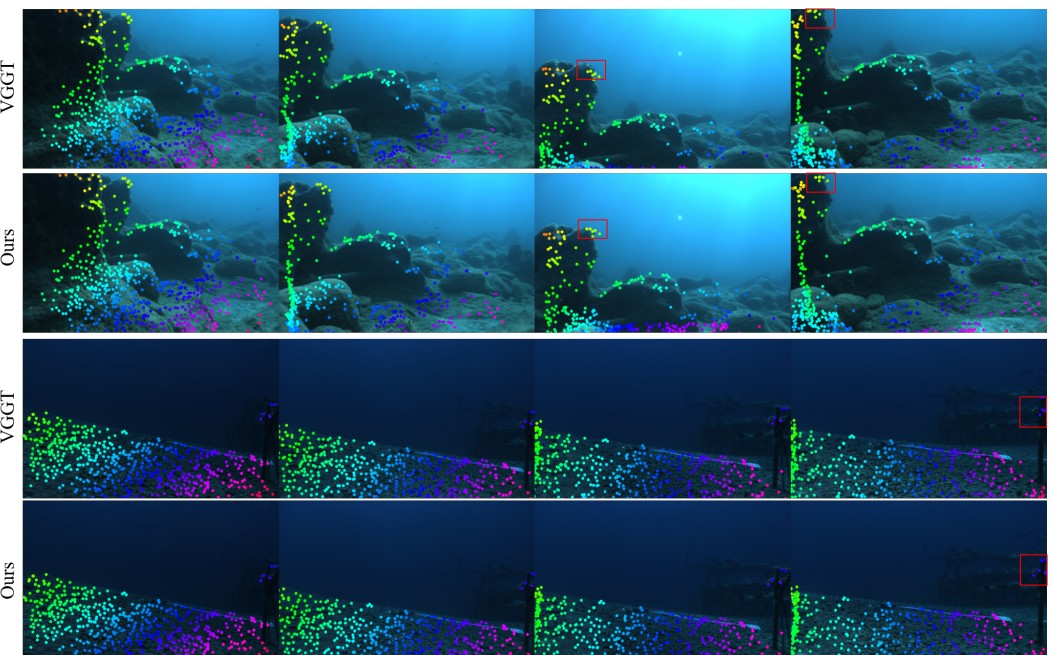

Figure 18: Qualitative multi-view point tracking with VGGT.

