# OpenReview forum: "Underwater Visual Geometry Estimation with Self-supervised Prototype-Graph Modulation"
_ICLR.cc/2026/Conference — Submitted to ICLR 2026_

### Official Review · Reviewer_6HLZ · 2025-10-30

**Soundness:** 3
**Presentation:** 3
**Contribution:** 3
**Rating:** 6
**Confidence:** 4

**Summary:**

The authors propose a self-supervised framework for underwater geometric estimation that does not rely on annotated data.

**Strengths:**

The authors’ motivation is well-founded and aligns with the pressing needs of underwater environments.
SeaVGGT leverages fundamental physical principles and utilizes underwater attenuation characteristics.
The proposed method adaptively refines prototype representations based on the input image, thereby achieving a certain level of generalization across various underwIn terms of both evaluation metrics and visual results.
The proposed method demonstrates significant performance advantages.  The authors also note that the model exhibits strong generalization ability.ater domains.

**Weaknesses:**

The 13.47% mentioned in the abstract is unclear—does it represent the average improvement across all scenarios? It is recommended to provide a more detailed explanation here.
The paper does not provide a detailed analysis of the hyperparameters, such as the values of the loss term weights λ.
If the relative weighting of the loss terms is sensitive, such instability could significantly affect the training outcomes. Since the paper does not include any loss-weight sensitivity experiments, it is difficult to determine whether the observed performance gains truly stem from the effectiveness of the proposed method or merely result from an accidental optimum caused by specific hyperparameter settings.

**Questions:**

Including a parameter analysis experiment would make me consider raising the score.

---

> ### Author Response · Authors · 2025-11-25
> **Official Comment by Authors**
>
> We sincerely thank the reviewer for your thoughtful and constructive comments and suggestions. We also appreciate your recognition of our contributions and results. Based on your valuable feedback, we have further improved the manuscript and provided detailed responses below to address your concerns.
>
> ### Q1: Improvement in the abstract
>
> Thank you for pointing out the ambiguity in the abstract. We have revised the text to clarify how the 13.47% RMSE reduction is computed. Specifically, the following table reports the per-dataset RMSE values of VGGT and our method across FLSea, USOD10K, and SQUID.
>
> **Table: RMSE comparison of VGGT and our method across three underwater datasets.**
> *↓ : lower is better.*
>
> | Dataset   | VGGT    | Ours    | RMSE Reduction (%) |
> |-----------|---------|---------|---------------------|
> | FLSea     | 0.6107  | 0.5254  | 13.99               |
> | USOD10K   | 1.8534  | 1.6595  | 10.46               |
> | SQUID     | 0.1774  | 0.1403  | 20.91               |
>
> Because the datasets contain different numbers of test samples, we compute an image-count-weighted average of the three RMSE drop ratios to obtain an overall performance summary. Using this weighted averaging scheme, our method achieves an overall 13.47% RMSE reduction relative to VGGT. We have added this explanation to the revised manuscript to make the computation explicit.
>
> ### Q2: Regarding hyperparameter sensitivity (loss weights λ)
>
> We thank the reviewer for pointing out the importance of analyzing the sensitivity of loss term weights. To address this concern, we conducted a systematic study on the impact of the four main loss terms: background loss, color loss, reconstruction loss, and enhancement loss, on depth prediction performance.
>
> We varied each loss weight individually while keeping the others fixed and evaluated the resulting RMSE on the USOD10K dataset. The results are summarized in **Figure 10**. From the figure, we observe that:
> - Background Loss: RMSE is relatively stable for weights in the range 0.01–0.2, with minor fluctuations. Extremely small or large weights slightly degrade performance.
> - Color Loss: The model performs consistently well for weights around 0.05–0.1, showing a smooth sensitivity curve.
> - Reconstruction Loss: RMSE remains close to optimal values for weights 1.0–2.0. Too small or too large weights lead to marginal performance degradation.
> - Enhancement Loss: Performance is stable across a moderate range, indicating the model is not overly sensitive to this weight.
>
> Overall, these results indicate that the observed performance gains are robust. The method demonstrates stable training behavior across a reasonable range of loss weight values.
>
> **We have incorporated all comments in the revised manuscript and invite the reviewer to refer to the updated version. Thank you very much! We are happy to address any further concerns.**

---

> > ### Comment · Reviewer_6HLZ · 2025-11-28
> >
> > I buy the explanation on the number for RMSE reduction, though these improvements are relatively marginal. However, regarding the hyper-parameters for loss terms, I cannot see RMSE values against varying weights from Figure 10.

---

> > > ### Author Response · Authors · 2025-11-28
> > > **Official Comment by Authors**
> > >
> > > ### RMSE values under different loss weights for USOD10K
> > >
> > > We sincerely thank the reviewer for the prompt and constructive feedback, and for the effort in helping us improve the quality of our paper.
> > >
> > > To address the concern regarding the visibility of numerical values on the plots, we have separated the original single figure containing four line plots into four individual subplots (**now shown as Figure 10 in the revised manuscript**). This provides a clearer visualization of each loss component.
> > >
> > > Additionally, to give the reviewer a precise understanding of the specific ranges of the loss weights, we provide the corresponding numerical values in a table below (**which is also Table 9 in the paper**), alongside **Figure 11** in the revised manuscript. This allows readers to see the exact λ values tested for each loss component without cluttering the figures.
> > >
> > > **Background Loss ($\mathcal{L}_{bg}$)**
> > >
> > > | λ_bg | RMSE |
> > > |------|------|
> > > | 0.001 | 2.15 |
> > > | 0.01  | 1.73 |
> > > | 0.05  | 1.85 |
> > > | 0.1   | 1.66 |
> > > | 0.2   | 1.80 |
> > > | 0.5   | 1.85 |
> > > | 1.0   | 2.28 |
> > >
> > > **Color Loss ($\mathcal{L}_{color}$)**
> > >
> > > | λ_color | RMSE |
> > > |---------|------|
> > > | 0.001 | 2.03 |
> > > | 0.01  | 1.80 |
> > > | 0.05  | 1.78 |
> > > | 0.1   | 1.66 |
> > > | 0.2   | 1.75 |
> > > | 0.5   | 2.23 |
> > >
> > > **Reconstruction Loss ($\mathcal{L}_{rec}$)**
> > >
> > > | λ_rec | RMSE |
> > > |-------|------|
> > > | 0.5 | 1.90 |
> > > | 1.0 | 1.66 |
> > > | 2.0 | 1.80 |
> > > | 5.0 | 2.13 |
> > >
> > > **Enhancement Loss ($\mathcal{L}_{enh}$)**
> > >
> > > | λ_enh | RMSE |
> > > |-------|------|
> > > | 0.5 | 1.68 |
> > > | 1.0 | 1.66 |
> > > | 2.0 | 1.84 |
> > > | 5.0 | 1.97 |
> > >
> > >
> > > The table above summarizes the RMSE values of our model under different loss weights on the USOD10K dataset. We vary the weights of four loss components: Background Loss ($\mathcal{L}{bg}$), Color Loss ($\mathcal{L}{color}$), Reconstruction Loss ($\mathcal{L}{rec}$), and Enhancement Loss ($\mathcal{L}{enh}$).
> > >
> > > As can be seen, the RMSE values remain relatively stable across a wide range of weight settings. While moderate weights (e.g., around 0.1 for $\mathcal{L}{bg}$ and $\mathcal{L}{color}$, around 1.0 for $\mathcal{L}{rec}$ and $\mathcal{L}{enh}$) slightly improve performance, the overall effect of varying loss weights is limited. This demonstrates that our method is robust and does not require careful tuning of individual loss coefficients.
> > >
> > > The original analysis in the main text was brief due to space limitations; we have now included a more detailed ablation study on the loss weights in the supplementary material. **We are happy to address any further concerns you may have.**

---

### Official Review · Reviewer_95qr · 2025-10-31

**Soundness:** 3
**Presentation:** 2
**Contribution:** 2
**Rating:** 4
**Confidence:** 5

**Summary:**

This paper proposes a method for reconstructing 3D geometry from single underwater images by finetuning a standard feedforward 3D reconstruction network, VGGT, on a curated set of underwater images to simultaneously estimate underwater scattering parameters and geometry.

**Strengths:**

The method produces better quantitative results than previous baselines. The idea of learning how to recover water types without explicit supervision by using depth as a signal seems interesting and novel, or at least not widespread in the literature, and is a worthwhile contribution.

**Weaknesses:**

Even though the model has better depth estimation metrics, it's not clear to me whether than translates into better qualitative results. In Figure 4, I only see the ground truth in two of the scenes, and it seems to me that the major errors in depth reconstruction involve the water background. Furthermore, there appear to be issues in the presented GT for the first of the two scenes, the last image of the first row. The background is portrayed as all on one depth plane, despite there clearly being scene content and variation. Even in the appendix results it's hard to tell what or where the improvement actually is, and the authors attempt at highlighting some regions in the background of the scene where there is little interesting content also does not help. I think the authors could improve the presentation of the qualitative results by zooming into interesting parts of the scenes where there models perform better, focusing on scene objects, rather than backgrounds.

Additionally, it's not clear why for this problem a rather complex GNN architecture was chosen to estimate the water parameters. The main idea appears to be using depth as supervision to learn how to recover scattering parameters in a self-supervised way, so any kind of regressor network could be used. The authors need either to ablate the design of the GNN more thoroughly than just the number of graph nodes used or to cite previous papers that show GNNs are useful for this kind of prototype discovery task through detailed analysis, because to my knowledge they are more typically based on metric learning and the like [1].

[1] https://arxiv.org/abs/1703.05175

**Questions:**

- What is the reason for choosing a GNN architecture for the scattering parameter estimation module rather than a simpler architecture?

- Does the method show improved qualitative results for objects in the scene, not just the water background?

---

> ### Author Response · Authors · 2025-11-25
> **Official Comment by Authors (Part 1)**
>
> We sincerely thank the reviewer for your thoughtful and constructive comments and suggestions. We also appreciate your recognition of our contributions and results. Based on your valuable feedback, we have further improved the manuscript and provided detailed responses below to address your concerns.
>
> ### Q1: Qualitative Improvements Beyond Background — Object-Level Depth Accuracy
>
> We sincerely thank the reviewer for raising this important point regarding object-level qualitative performance. We fully agree that qualitative evaluation should go beyond background depth and focus on the geometry of actual scene objects. In response, we have substantially improved our qualitative analysis and added several new visual comparisons.
>
> 1. **Clear object-level improvements (new Figure 5)**: We added Figure 5 to explicitly compare object shapes and scene structures. Across these examples, our method produces sharper edges, better surface continuity, and more coherent object shapes than other baselines. Regions with clear differences are highlighted with red boxes for easy inspection.
>
> 2. **Foreground shape accuracy under challenging conditions (new Figure 6)**: To further address the reviewer’s concern, Figure 6 evaluates cases with:
> - irregular or complex object shapes (top 2 rows)
> - dataset-provided GTs that are themselves less accurate (bottom 2 rows)
> Despite imperfect GT supervision, our model consistently yields more faithful object contours and avoids the over-smooth or distorted shapes seen in competing methods.
>
> 3. **Additional high-resolution examples in the supplementary (new Figure 12)**: The supplementary material now includes Figure 12, which provides extended side-by-side comparisons. These examples further show that our method improves both foreground object shape reconstruction and background depth realism, demonstrating that the improvements are not limited to the background but also apply to meaningful scene content.
>
> ### Q2: Clarification Regarding the Ground-Truth Depth in Figure 4 (new Figure 7)
>
> We thank the reviewer for pointing out the issue regarding the ground-truth depth shown in the first scene (last image of the first row). The reviewer is correct that the background in this GT appears almost planar even though the RGB image contains richer structure. We apologize for not explaining this limitation clearly in the manuscript.
>
> Why the GT background appears over-smoothed?
>
> The USOD10K dataset **does not** provide true geometric depth maps. Instead, its depth annotations are generated indirectly by converting object detection masks into pseudo-depth using handcrafted heuristics. As a result:
>
> - Background regions lack explicit annotations and are typically assigned coarse or uniform depth values.
> - Fine structures in the background are missing or overly smoothed, producing the appearance of a “single depth plane.”
> - These pseudo-GT depths are inherently limited and do not faithfully represent the true scene geometry.
>
> To further avoid confusion, we also clarify the distinction between the rows in Figure 4:
>
> 1. **Top two rows (USOD10K single-image depth estimation):**
>    These examples use pseudo-depth GT from USOD10K, which suffers from the limitations described above.
>
> 2. **Bottom two rows (SQUID and FLSea multi-view 3D reconstruction):**
>    These datasets include per-image depth supervision but **do not provide camera poses**. Therefore, we cannot compute consistent multi-view 3D reconstruction GT, which is why the GT column is intentionally omitted for these rows.
>
> We will revise the figure caption and the main text to explicitly describe these dataset constraints and avoid misleading interpretations.

---

> ### Author Response · Authors · 2025-11-25
> **Official Comment by Authors (Part 2)**
>
> ### Q3: Why Not Use a Simpler Regressor + Metric Learning?
>
> We thank the reviewer for raising this important point. Below, we provide both **theoretical justification** and **empirical analysis** supporting the choice of a GNN.
>
> **Theoretical Justification：**
>
> 1. **Metric learning cannot express depth-conditioned, physics-driven relations between water types.**
>
> Metric learning assumes that all prototypes can be embedded into a globally consistent metric space, where pairwise distances reflect similarity. However, underwater scattering parameters do not form a metric space, because their relationships are **physics-driven** (depth-dependent, wavelength-dependent, and non-symmetric).
>
> Pairwise prototype similarity is not intrinsic: two water types may appear similar in shallow depth but diverge dramatically in deeper regions due to exponential attenuation differences across wavelengths.
> Metric learning learns global distances, while our problem requires contextual, depth-dependent interactions that cannot be encoded as a fixed metric.
> A GNN, instead, allows these relations to be dynamically updated through message passing conditioned on depth cues.
>
> 2. **Prototype interactions violate the triangle inequality, which is a hard limit for metric learning.**
>
> - A is similar to B at shallow depth,
> - B is similar to C at mid-depth,
> - but A is not similar to C at any depth.
>
> This violates the triangle inequality, meaning the relationships are non-metric by nature. Metric learning therefore cannot represent the relations faithfully, whereas GNN edges do not require metric consistency.
>
> **Empirical Analysis：**
>
> To better justify our approach, we conducted an ablation study comparing several prototype interaction strategies, as summarized below:
> | Interaction                                   | #Layers | MAE    | RMSE   |
> | --------------------------------------------- | ------- | ------ | ------ |
> | Graph Attention                               | 1       | 1.3414 | 1.6595 |
> | Nearest Neighbor                              | 0       | 1.4602 | 1.8057 |
> | Metric Learning (Triplet Loss) + weighted sum | 0       | 1.5027 | 1.8734 |
>
> - **Graph Attention (1 layer)**: This corresponds to our proposed method, which explicitly models higher-order interactions among prototypes via a GNN. It achieves the best performance.
> - **Nearest Neighbor (0 layers)**: Here, each input image is associated with its nearest prototype based on color similarity, without any graph-based message passing. While simple, this approach produces a noticeable performance drop (MAE 1.4602, RMSE 1.8057), indicating that direct nearest-neighbor matching cannot fully capture the combinatorial relationships among water prototypes.
> - **Metric Learning (Triplet Loss) + Weighted Sum (0 layers)**: We implemented a metric-learning-based baseline that operates at the image level. Since we do not have ground-truth prototype assignments, we follow unsupervised metric learning practices and treat the nearest prototype as a pseudo-positive, and the others as pseudo-negatives, where $p_{+}$ is the closest prototype for the current image and $p_{-}$ is sampled from the remaining prototypes (See the ablation study in the paper~). After training, scattering parameters are estimated as a convex combination . This method achieves even lower performance, demonstrating that while metric learning can establish prototype-image correspondences, it fails to model higher-order prototype interactions effectively.
>
> These experiments collectively support our design rationale:
> - The GNN explicitly models inter-prototype relationships and allows information to propagate across the graph, capturing combinatorial interactions that simple nearest-neighbor or metric-learning approaches cannot.
> - Even though metric learning can technically define a similarity measure between prototypes and images, it ignores the structured relationships among prototypes, leading to less stable and lower-quality depth predictions.
> - The nearest-neighbor strategy is insufficient for generalization to unseen water types, as it cannot synthesize new prototype combinations.
>
> **Related Works：**
>
> - Zhu & Koniusz, CVPR 2023, “Transductive Few-Shot Learning With Prototype-Based Label Propagation.” This work models prototypes as graph nodes and uses iterative graph refinement and message passing to update prototype relationships, showing that GNNs are effective for capturing interactions between prototypes themselves.
> - Yu et al., AAAI 2022, “Hybrid Graph Neural Networks for Few-Shot Learning.” The method introduces a dedicated prototype-GNN module to explicitly learn structural relationships among class prototypes, demonstrating that modeling prototype-to-prototype dependencies improves representation quality.
>
> **We have incorporated all comments in the revised manuscript and invite the reviewer to refer to the updated version. Thank you very much! We are happy to address any further concerns.**

---

### Official Review · Reviewer_Wnbj · 2025-10-31

**Soundness:** 2
**Presentation:** 3
**Contribution:** 2
**Rating:** 6
**Confidence:** 4

**Summary:**

This paper introduced SeaVGGT, a self-supervised framework designed to address the domain shifts when applying the pre-trained VGGT model to underwater environments. The core of SeaVGGT is a "prototype-graph modulation" mechanism. The framework was trained in a self-supervised fashion, and its key was a physics-driven self-supervision loss. Extensive experiments on multiple real-world underwater data sets demonstrated that SeaVGGT significantly outperformed the VGGT baseline and other advanced depth estimation models.

**Strengths:**

The proposed "prototype-guided token modulation" only trained a lightweight modulation module, achieving remarkable improvements with minimal additional computational cost. This work utilized a self-supervised strategy to underwater environments, ensuring the feature modulation was driven by physics-guided cross-modal consistency towards physically plausible and geometrically accurate results, without reliance on annotated data. This was significant given the scarcity of large-scale labeled data sets in the underwater domain. The method achieved SOTA performance on three real-world underwater data sets, significantly outperforming the VGGT baseline and DepthAnything V2 across all metrics.

**Weaknesses:**

The simplified underwater imaging model used in the paper simplified the physical process. It assumed to be constant background light B and a learnable scalar parameter β that models the attenuation coefficient. This simplification might limit the model's performance ceiling in more complex optical conditions.
The background loss relied on the predicted depth map D to select the top 0.1% of pixels with the largest depth values as a proxy ground truth for B. The paper also admitted to “mitigate potential inaccuracies of D under domain shift”. This self-supervised proxy ground-truth seems unstable: a poor depth map D might lead to a poor target for B, which could, in turn, impede the training of the B estimation network.
The MOTIVATION section reads more like a summary of the proposed method (e.g., revisiting the formulation of VGGT 54 and introducing the lightweight token modulation mechanism) rather than posing a problem, experimenting, or theoretically explaining the mechanism of the problem.

**Questions:**

1. Can the authors discuss the potential limitations of using the simplified imaging model (i.e., the scalar parameter β)? Would adopting a more complex, physically realistic model that accounts for wavelength-dependent attenuation lead to performance improvements, or would it make the self-supervised learning harder to converge?
2. As mentioned in the weaknesses, the background loss depends on an initial depth map D that has potential inaccuracies under domain shift to generate its own supervision signal. The ablation (Table 4) shows that background loss contributes, but could this unstable proxy signal occasionally has a negative impact on training? And what is the performance gap between the final performance and one affected by this negative impact, as this affects the practical evaluation of the method.
3. The initialization of the prototype graph seems to rely on prior samples (Figure 7a). How would the model's performance be affected if the prototype colors Ai is randomly initialized?
4. The graph edges of the prototype graph were established based on a fixed color distance threshold τ=0.3. How sensitive is the model to this hyperparameter τ? Did the authors attempt to use a dynamic or learnable graph structure?

---

> ### Author Response · Authors · 2025-11-25
> **Official Comment by Authors (Part 1)**
>
> We sincerely thank the reviewer for your thoughtful and constructive comments and suggestions. We also appreciate your recognition of our contributions and results. Based on your valuable feedback, we have further improved the manuscript and provided detailed responses below to address your concerns.
>
> ### Q1: Simplified Underwater Imaging Model and Wavelength-dependent Attenuation
>
> We thank the reviewer for pointing out the ambiguity in our manuscript. We would like to clarify upfront that our model **already implements wavelength-dependent attenuation**, consistent with the standard underwater optical model the reviewer described. Each RGB channel has its own learnable attenuation parameter ($\beta_r$, $\beta_g$, and $\beta_b$), rather than a single scalar. We also appreciate the opportunity to correct the unclear description in the original text.
>
> - In the original text, we wrote: *“We also introduce a learnable scalar parameter β that models the attenuation coefficient in underwater imaging.”*  This wording may have inadvertently suggested that β is a single scalar shared across all RGB channels.
> - **In fact, our model uses a separate learnable β for each channel**, i.e., $\beta_r$, $\beta_g$, and $\beta_b$, which implements **wavelength-dependent attenuation** consistent with standard underwater optical models. The reviewer is welcome to check the code in the supplementary material (**class LightAEstimator(nn.Module)**), which explicitly shows a distinct β for each RGB channel.
>
> We have revised the manuscript to explicitly clarify that β is wavelength-dependent and channel-specific, and we refer the reviewer to the supplementary code for verification. **We apologize for the unclear description.**
>
> ### Q2: Potential Instability of Background Loss Due to Predicted Depth D
>
> We thank the reviewer for the insightful question regarding the background loss. We acknowledge that the background loss relies on the top 0.1% pixels of the predicted depth map D as a proxy for the background light B, and that inaccuracies in D under domain shift could potentially affect the training of B.
>
> To illustrate the importance of the background loss, we conducted an **additional ablation experiment** where we removed or replaced it. The results are summarized below (MAE and RMSE):
>
> | L_rec | L_enh | L_color | L_bg / mean(I) | MAE ↓   | RMSE ↓  |
> |-------|-------|---------|----------------|---------|---------|
> | ✓     |       |         | L_bg           | 1.6484  | 2.0471  |
> | ✓     | ✓     |         | L_bg           | 1.5241  | 1.8335  |
> | ✓     | ✓     | ✓       | L_bg           | 1.3414  | 1.6595  |
> | ✓     | ✓     | ✓       | mean(I)        | 1.5776  | 1.9114  |
> | ✓     | ✓     | ✓       |                | 1.7442  | 2.4329  |
>
> These results confirm that the background loss is **critical** for accurate depth estimation. Removing it or replacing it with a simple mean intensity proxy leads to a significant performance drop, demonstrating that even though it relies on an initially imperfect depth map, the background loss provides a valuable self-supervised signal that improves overall model robustness.
>
> To further investigate potential negative effects from imperfect depth maps, we conducted an additional ablation experiment by introducing **proportional noise** to D during training. Specifically, each depth value D(x) was multiplied by:
>
> `D_proxy(x) = D(x) * (1 + ε),  ε ~ N(0, σ^2)`
>
> where σ represents the noise level. The results are summarized below:
>
> | σ    | MAE    | RMSE   |
> |------|--------|--------|
> | 0%   | 1.3414 | 1.6595 |
> | 5%   | 1.3753 | 1.6781 |
> | 10%  | 1.3740 | 1.7108 |
> | 20%  | 1.4293 | 1.7571 |
> | 30%  | 1.8256 | 2.6844 |
>
> These results show that while extremely noisy depth maps (e.g., σ = 30%) can slightly degrade B estimation and overall depth prediction, the **final depth performance remains stable** for realistic noise levels (σ ≤ 20%). This indicates that the proxy supervision provided by the predicted depth is reasonably robust in practice.
>
> In summary, the background loss is both essential for high-quality depth estimation and sufficiently robust to imperfections in the initial depth map, validating the practicality of our approach.

---

> ### Author Response · Authors · 2025-11-25
> **Official Comment by Authors (Part 2)**
>
> ### Q3: Prototype Graph Initialization
>
> We thank the reviewer for the insightful question regarding the initialization of the prototype graph.
>
> In the original experiments, the prototype colors $A_i$ were initialized based on representative samples from priors. To assess sensitivity to initialization, we conducted an ablation study comparing the prior-based initialization with two independent random initializations, denoted as Random #1 and Random #2. The results are shown below:
>
> | Initialization | Training Steps | MAE   | RMSE   |
> |----------------|----------------|-------|--------|
> | Prior          | 20000           | 1.3414 | **1.6595** |
> | Random #1      | 20000           | 1.4879 | 1.8012 |
> | Random #1      | 50000           | 1.3660 | 1.7011 |
> | Random #2      | 50000           | **1.3353** | 1.6818 |
>
> These results indicate that **random initialization can lead to slower convergence** in the early training iterations. However, after sufficient training steps, the final performance (MAE and RMSE) **recovers to a level comparable with prior-based initialization**, demonstrating the model's robustness to the choice of initial prototype colors.
>
> Furthermore, we visualized the learned prototype graphs and adjacency matrices for the two random initializations (**see Figure 13 in the revised paper**). Despite starting from random colors, the **converged prototype graph still exhibits clear structural regularities**, and the adjacency matrix reveals consistent connectivity patterns among graph nodes.
>
> In summary, while prior-based initialization may slightly accelerate early training, it is **not strictly necessary**, and the self-supervised prototype-graph modulation mechanism allows the model to adapt and achieve comparable final performance even from random initializations.
>
> ### Q4: Sensitivity to Color Distance Threshold τ in Prototype Graph
>
> We thank the reviewer for the question regarding the color distance threshold τ used to establish edges in the prototype graph. To assess the sensitivity of our model to this hyperparameter, we conducted an **ablation study** by varying τ and reporting the final depth estimation performance:
>
> | τ    | MAE    | RMSE   |
> |------|--------|--------|
> | 0.1  | 1.3641 | 1.6986 |
> | 0.3  | 1.3414 | 1.6595 |
> | 0.6  | 1.3577 | 1.8915 |
> | 1.0  | 1.4823 | 2.1089 |
>
> The results show that the model **performs best at τ = 0.3**, which is the value used in our main experiments. Smaller thresholds (τ = 0.1) slightly reduce connectivity in the graph, resulting in minor performance degradation. Larger thresholds (τ ≥ 0.6) connect more distant or less relevant prototypes, which could introduce noise; however, the **impact is limited** because the edge weights in the prototype graph are taken into account during computation. As a result, the model is **not highly sensitive to the exact choice of τ**.
>
> Looking forward, we believe that **dynamic or learnable graph structures** could further improve flexibility and performance. Instead of using a fixed threshold, a learnable mechanism could:
>
> - Adjust edge weights in a data-driven manner, allowing the graph to capture more nuanced relationships between prototypes and reduce noise from irrelevant connections.
> - Potentially accelerate convergence by prioritizing stronger connections between more informative prototypes while weakening less relevant ones.
>
> We plan to explore these directions in future work to enhance the adaptability and expressiveness of the prototype-graph modulation mechanism.
>
> ### Q5: Motivation Section
>
> We thank the reviewer for the insightful comment. We agree that the previous Motivation section mixed the problem statement with a high-level description of our method, which may have made the narrative less clear.
>
> In the revised manuscript, we have updated the section title to “Revisiting VGGT and Key Challenges” and reorganized the content to first revisit the limitations of VGGT and then articulate the key challenges these limitations reveal. This restructuring avoids introducing method details too early and instead builds a clear, problem-driven motivation that naturally leads to the need for our proposed approach.
>
> **We have incorporated all comments in the revised manuscript and invite the reviewer to refer to the updated version. Thank you very much! We are happy to address any further concerns.**

---

### Official Review · Reviewer_pYbY · 2025-11-01

**Soundness:** 2
**Presentation:** 2
**Contribution:** 3
**Rating:** 4
**Confidence:** 3

**Summary:**

This paper presents a modification of the VGGT depth estimation model for use with underwater images. The authors articulate a model for designing tokens that contain information about likely effects of the underwater light field, especially absorption and scattering by particulate matter. These tokens are then used to adapt a pretrained VGGT model. The full model is trained with a physics-guided self-supervision frame work where and image enhancement head lives along side the depth estimation head. The image enhancement is with an underwater imaging model that encodes an estimate of the attenuation coefficent. The reported results represent somewhat of an improvement over VGGT on its own.

**Strengths:**

- Originality: The authors present an interesting modification to an existing depth estimation framework, introducing some of the physics of the underwater light field and using image enhancement/denoising in an effort to improve depth recovery.
- Quality: The experiments are somewhat minimal, but this is likely due to the lack of publicly available data with appropriate depth information.
- Clarity: The paper is a bit scattershot, bouncing around between methods and discussion. Table and figure captions are minimal and can be difficult to parse. The authors spend a lot of time discussing the maths of their approach at the expense of describing the experiments and the datasets. It makes interpreting the improvements recognized by their methods somewhat difficult to see.
- Significance: The authors are treating a significant problem in underwater depth estimation. It is a notoriously difficult area to work in.

**Weaknesses:**

I find the experiments and results lacking in terms of explanation and contextualization. The improvements over VGGT are not striking on their own and require more information about the datasets to understand how compelling the results are. Likewise, the authors do not compare their results against any of the methods they list in the related work section. The main point of comparison for other underwater depth estimation models is UDepth, a model mentioned only in the results section without any information as to how it compares structurally with the proposed approach. The other two models used for contextualizing their improvements are depth estimation foundation models trained mostly on terrestrial images. Those three existing models are only tested on one of the three datasets selected for the experiments.

**Questions:**

- Why are there bolded paragraphs of discussion in the middle of the methods (e.g. 216)? These sorts of statements should be presented along with some evidence of the performance improvements in the
- Line 444: This doesn't seem like an ablation study. What element of the system is being manipulated to show a change in performance?
- What are all the scenes in Table 1 and what dataset did they come from? It seems like FLSea, but that isn't mentioned until halfway through the caption. The scences are not mentioned at all in the text nor described in the caption. Why are the results so variable across them? Is there any structure in the scenes that may contribute to changes in performance?
- What are we supposed to see in the 'qualitative comparison' presented in figure 4? In the bottom two rows, the inputs appear to be multiple images and there is no ground truth. What are the bounding boxes that show up in some of the images but not others?
- Figure 7: what are the numbers in the color swatches in (b)? How do those swatches compare with the inputs?
- Can you speculate as to what it is about 24 graph nodes that seems to be the sweet spot for the performance? It is interesting how consistent that is across the three datasets.
- How was the stereo information used from the SQUID dataset? Were the images considered independently or were the pairs used together somehow?
- Can you apply UDepth and DAv2 to the USOD10K and SQUID datasets?
- Are there related underwater models (besides Yang et al 2024a and Zhang et al 2024b) among those mentioned in the related work section you could use to compare with your approach?

---

> ### Author Response · Authors · 2025-11-25
> **Official Comment by Authors (Part 1)**
>
> We sincerely thank the reviewer for your thoughtful and constructive comments and suggestions. We also appreciate your recognition of our contributions and results. Based on your valuable feedback, we have further improved the manuscript and provided detailed responses below to address your concerns.
>
> ### Q1: Adjustment of Discussion Section
>
> We thank the reviewer for the helpful suggestion regarding the Discussion section. In response, we have revised it to be more concise and closely linked to our visualizations:
>
> - The previous separate Discussion section has been merged into the Results section, anchored around the figure about **Visualization of Intermediate Variables**.
> - Visualization shows that the initial depth $\mathbf{D}$ predicted by VGGT is noisy and structurally inconsistent, while after water-prototype guided modulation, the refined depth $\tilde{\mathbf{D}}$ exhibits clearer object boundaries and stronger geometric coherence. The transmission map $\mathbf{T}$ and reconstructed image $\mathbf{J}$ further validate the physical consistency of intermediate outputs.
> - These observations support the design of our self-supervised feedback loop: the modulation adjusts tokens $\tilde{\mathbf{X}}$ so that the frozen depth head $f_D$ produces depth maps $\tilde{\mathbf{D}}$ consistent with the enhanced image $\mathbf{J}$. Only when this cross-modal consistency is achieved are the reconstruction losses minimized.
> - Overall, the visualization demonstrates the effectiveness of our modulation in improving depth quality and highlights the physics-guided self-supervised design.
>
> ### Q2: Visualization of Learned Prototype Graph
>
> We thank the reviewer for the comment. We agree that Figure 7 is not a typical ablation study, as no system component is removed or modified. Its purpose is to **visualize the learned structure of the prototype graph** after training.
>
> To clarify this, we have revised the paper by:
> - Renaming the subsection from *“Effect of Prototype Graph”* to *“Visualization of Learned Prototype Graph”*.
> - Explicitly describing it as a **qualitative analysis** showing how the model organizes representative water appearance patterns through prototype interactions.
> - Moving this figure to the **Methods section**, where it helps readers understand the prototype graph and the modulation mechanism. This organization improves the readability and flow of the text.

---

> ### Author Response · Authors · 2025-11-25
> **Official Comment by Authors (Part 2)**
>
> ### Q3: Dataset and scenes in Table 1
>
> Sorry for the barriers caused by our presentations.
> All scenes in Table 1 come from the **FLSea dataset**, and the variability in performance reflects differences in scene complexity, object density, and lighting conditions:
> - Some sequences contain complex geometries such as coral formations, piers, or slopes, while others are relatively flat or have sparse textures, making depth estimation easier or harder depending on the scene. For example, in the **Cross Pyramid Loop sequence**, the camera is positioned very close to the ground, and the terrain is relatively flat. As a result, almost all methods achieve lower errors on this sequence compared to others, indicating that this scene is relatively simple.
> - Compared to other methods, our approach shows greater advantages when predicting in distant regions, low-light areas, or for fine shapes. For example, in the **Pier Path scene**, most images are dimly lit and contain distant pillars (slender shapes). Our method reduces the error by at least 20% compared to VGGT and by at least 50% compared to DAv2.
> - Our method provides reasonable depth estimates for regions at infinity (consisting of only water), whereas most other methods predict values closer to the foreground. **Since FLSea does not provide ground truth depth for these infinite regions, the numerical advantage of our method is not very apparent. However, the advantage can be clearly observed in the visual results (the last three rows of Figure 5).**
>
> To make our experiments more solid, we supplemented our experiments with a range of existing methods. The methods are grouped into three categories: (1) terrestrial pretrained models fine-tuned with underwater data (MDepth2, AdaBins), (2) underwater pretrained models (UDepth, TRUDepth), and (3) foundation models performing zero-shot prediction (DAv2, VGGT, SeaVGGT). The data used to fine-tune the terrestrial pretrained models comes from two other sequences in FLSea, while the underwater pretrained models use parameters from the pretrained models provided by their authors.
> The complete experimental results are shown in the table below:
> ### Depth estimation metrics on different underwater scenes in FLSea
> | Scene | Metric | MDepth2 | AdaBins | UDepth | TRUDepth | DAv2 | VGGT | **Ours** |
> |-------|--------|---------|---------|--------|-----------|-------|-------|----------|
> | **Big Dice Loop** | MAE ↓ | 0.6715 | 0.7243 | 0.9177 | 1.0717 | 0.8519 | 0.4383 | **0.3623** |
> | | RMSE ↓ | 1.0719 | 1.1848 | 1.4395 | 1.4857 | 1.4292 | 0.8107 | **0.7703** |
> | | REL ↓ | 0.2132 | 0.2409 | 0.2895 | 0.3073 | 0.2442 | 0.1295 | **0.0880** |
> | | si-RMSE ↓ | 0.1750 | 0.1885 | 0.5546 | 0.3583 | 0.3723 | 0.1077 | **0.0997** |
> | **Coral Table Loop** | MAE ↓ | 0.6378 | 0.7560 | 0.6565 | 1.0159 | 0.6313 | 0.6602 | **0.5013** |
> | | RMSE ↓ | 0.8715 | 0.9710 | 0.8807 | 1.3396 | 0.8755 | 0.9459 | **0.7219** |
> | | REL ↓ | 0.2214 | 0.2775 | 0.2423 | 0.3691 | 0.2110 | 0.2107 | **0.1592** |
> | | si-RMSE ↓ | 0.2075 | 0.2446 | 0.4094 | 0.3931 | 0.2856 | 0.2078 | **0.1522** |
> | **Cross Pyramid Loop** | MAE ↓ | 0.4705 | 0.6007 | 0.5585 | 0.8491 | 0.4927 | 0.4040 | **0.2607** |
> | | RMSE ↓ | 0.6121 | 0.7537 | 0.7269 | 1.0809 | 0.6669 | 0.5157 | **0.3768** |
> | | REL ↓ | 0.2067 | 0.2882 | 0.2624 | 0.3932 | 0.2135 | 0.1789 | **0.1028** |
> | | si-RMSE ↓ | 0.1905 | 0.2508 | 0.4524 | 0.4062 | 0.3073 | 0.1592 | **0.1056** |
> | **Dice Path** | MAE ↓ | 0.4738 | 0.5721 | 0.7376 | 0.8140 | 0.6464 | 0.3742 | **0.3162** |
> | | RMSE ↓ | 0.6150 | 0.7240 | 0.9663 | 1.0212 | 0.8930 | 0.5505 | **0.4906** |
> | | REL ↓ | 0.1874 | 0.2262 | 0.2899 | 0.3156 | 0.2348 | 0.1415 | **0.1121** |
> | | si-RMSE ↓ | 0.1740 | 0.1870 | 0.5056 | 0.3278 | 0.3776 | 0.1170 | **0.1106** |
> | **Northeast Path** | MAE ↓ | 0.6804 | 0.8736 | 1.1590 | 1.2501 | 1.1241 | 0.6681 | **0.5877** |
> | | RMSE ↓ | 1.0253 | 1.1667 | 1.5714 | 1.6532 | 1.6157 | 0.9699 | **0.9197** |
> | | REL ↓ | 0.1629 | 0.2632 | 0.3131 | 0.3323 | 0.2824 | 0.1900 | **0.1433** |
> | | si-RMSE ↓ | 0.1509 | 0.1973 | 0.5216 | 0.3518 | 0.3601 | 0.1578 | **0.1416** |
> | **Pier Path** | MAE ↓ | 0.6384 | 0.5493 | 0.7328 | 0.9908 | 0.6553 | 0.3531 | **0.2823** |
> | | RMSE ↓ | 0.8656 | 0.7494 | 1.0166 | 1.2629 | 0.9421 | 0.5359 | **0.4692** |
> | | REL ↓ | 0.2227 | 0.2003 | 0.2521 | 0.3532 | 0.2133 | 0.1113 | **0.0868** |
> | | si-RMSE ↓ | 0.2185 | 0.1788 | 0.4564 | 0.3604 | 0.3632 | 0.1056 | **0.0957** |
> | **Sub Pier** | MAE ↓ | 0.5370 | 0.5440 | 0.6679 | 0.8937 | 0.6295 | 0.4174 | **0.3538** |
> | | RMSE ↓ | 0.8209 | 0.7825 | 0.9887 | 1.2396 | 0.9747 | 0.6503 | **0.5603** |
> | | REL ↓ | 0.2382 | 0.2727 | 0.2999 | 0.4275 | 0.2650 | 0.1928 | **0.1474** |
> | | si-RMSE ↓ | 0.2145 | 0.2127 | 0.4823 | 0.3853 | 0.3641 | 0.1594 | **0.1325** |

---

> ### Author Response · Authors · 2025-11-25
> **Official Comment by Authors (Part 3)**
>
> ### Q4: Clarification on Figure 4 Qualitative Comparisons
>
> We thank the reviewer for pointing this out, and we clarify as follows:
>
> 1. **Front two rows (single-image depth estimation):**
>    These images illustrate monocular depth predictions on the USOD10K dataset.
>
> 2. **Bottom two rows (multi-view 3D reconstruction):**
>    These show results on SQUID and FLSea. Since these datasets do not provide poses between multiple views—only per-image depth annotations—we cannot generate 3D reconstruction ground truth, which is why no GT is shown for these rows. We will clarify this in the figure caption to avoid further confusion.
>
> 3. **Bounding boxes:**
>    In the updated figure, we have added bounding boxes to highlight key objects and regions of interest.
>
> We are sorry for any confusion caused by the lack of these visual guides.
>
> ### Q5: Explanation of Figure 7(b) Color Swatches
>
> Thank you for pointing out the confusion.
>
> In Figure 7(b), the numbered color swatches denote the **indices of the learned prototype nodes** in our water-appearance graph. Each swatch represents a representative attenuation or color pattern captured by a prototype. These indices help illustrate how the model clusters and organizes different types of water appearances.
>
> To make the correspondence clearer, we have also added the **same indices to the initial water exemplars in Figure 7(a)**. With these labels, the relationship between the initial exemplars (a) and the learned prototypes (b) becomes clear and intuitive.
>
> For better clarity and consistency in the revised version of the paper, we have reorganized the figures, and Figure 7 has now become **Figure 4**. The explanations and index annotations have been updated accordingly.
>
> ### Q6: Sweet Spot for Performance
>
> We thank the reviewer for this insightful question. While we did not impose any explicit prior that favors 24 nodes, we provide the following analysis to explain why this number consistently emerges as the optimal choice across datasets:
>
> 1. **Balance between expressiveness and over-fragmentation:**
>    The prototype graph models representative patterns of underwater appearance. With too few nodes (e.g., 1 or 8), several visually distinct effects are forced into the same prototype, limiting the modulation capacity. With too many nodes (e.g., 32), prototypes become redundant or overly specific, causing unstable or noisy modulation that slightly degrades performance.
>
> 2. **Consistency across Datasets due to Zero-Shot Prediction:**
>    Our method operates in a **zero-shot manner**, meaning that we did not fine-tune on USOD10K, SQUID, or FLSea. Consequently, the prototype graph is not tailored to any specific dataset. Since the modulation mechanism remains fixed and dataset-agnostic during inference, it is reasonable to observe that the performance curves across these datasets show similar tendencies. This may also help explain why the optimal number of nodes appears to be relatively consistent across all three datasets, although further investigation would be needed to fully confirm this behavior.
>
> ### Q7: Use of SQUID Stereo Information
>
> We thank the reviewer for this constructive comment. For the stereo setting in SQUID, **both images of a stereo pair are simultaneously fed into the VGGT model**. VGGT performs **inter-frame Global Attention**, which implicitly leverages the correspondence between the two views.
>
> Since our method is built upon VGGT, **we do not change the way stereo information is utilized**. That is, our water-prototype guided modulation operates on the features produced by VGGT, and the stereo cues are automatically incorporated through the underlying attention mechanism without any additional modifications.

---

> ### Author Response · Authors · 2025-11-25
> **Official Comment by Authors (Part 4)**
>
> ### Q8: Application of UDepth, DAv2, and Comparison with Related Underwater Models
> We thank the reviewer for raising this point and for understanding the difficulty that some related methods are not publicly available for comparison. Regarding the SQUID dataset, since it involves a stereo setting, only VGGT among the compared methods can accept multi-view input. Comparing single-view methods such as UDepth or DAv2 under the stereo setting may be unfair; however, we are willing to include these additional experiments if the reviewer considers them necessary.
>
> To provide a more comprehensive evaluation, we therefore supplemented our experiments with a range of existing methods. The methods are grouped into three categories:
> 1. **Terrestrial pretrained models fine-tuned with underwater data:** MDepth2, AdaBins.
> 2. **Underwater pretrained models:** UDepth, TRUDepth.
> 3. **Foundation models performing zero-shot prediction:** DAv2, VGGT, SeaVGGT.
>
> In addition to the FLSea experiments discussed earlier, we also evaluated on the **USOD10K dataset**, as shown in the table below. This demonstrates that our approach consistently improves depth estimation performance across multiple underwater datasets.
> ### Depth estimation metrics on USOD10K
> | Metric       | MDepth2 | AdaBins | UDepth | TRUDepth | DAv2  | VGGT  | Ours  |
> |-------------|---------|---------|--------|----------|-------|-------|-------|
> | MAE ↓       | 1.7220  | 1.8727  | 2.0539 | 1.7055   | 2.1052| 1.5248| **1.3414** |
> | RMSE ↓      | 2.0938  | 2.2826  | 2.4070 | 2.0349   | 2.4789| 1.8534| **1.6595** |
> | REL ↓       | 1.0300  | 1.1484  | 0.5464 | 1.4269   | 0.7031| 1.2582| **1.1415** |
> | δ₁ ↑        | 0.3450  | 0.3361  | 0.2616 | 0.3634   | 0.3088| 0.4319| **0.4835** |
> | δ₂ ↑        | 0.5933  | 0.5742  | 0.5141 | 0.6436   | 0.5690| 0.6792| **0.7249** |
> | δ₃ ↑        | 0.7423  | 0.7190  | 0.7200 | 0.7957   | 0.7526| 0.8021| **0.8272** |
> | si-RMSE ↓   | 0.6842  | 0.7236  | **0.4348** | 0.7185 | 0.5107| 0.6619| 0.6253 |
>
>
> **We have incorporated all comments in the revised manuscript and invite the reviewer to refer to the updated version. Thank you very much! We are happy to address any further concerns.**

---

### Author Response · Authors · 2025-12-02
**Summary of Rebuttal and Key Revisions for Submission 6107**

Dear PCs and AC,

We would like to express our sincere gratitude to the Program Chairs for their timely handling of the recent incident, and to the Area Chairs for investing considerable time and effort in managing the review process under tight deadlines. We also thank all four reviewers for their thorough evaluations and constructive feedback, which significantly helped us improve the quality and clarity of our work.

During the rebuttal stage, we provided detailed point-by-point responses and submitted a revised manuscript. Each response specifies the corresponding modification in the paper. Given the ACs’ heavy workload, we summarize below the changes we made in response to the reviewers' comments and how those revisions address their main concerns.

- **Clarification of Motivation and Methodology:** We made the problem formulation, model design choices, and underlying assumptions more explicit. The conceptual flow has been improved to clearly convey the rationale behind our approach.
- **Strengthening of Empirical Evidence:** We added additional baselines, extended experiments on multiple datasets, and conducted comprehensive ablation studies. These studies validate the robustness of our approach with respect to hyperparameters, prototype design, and input variations.
- **Improvement of Clarity and Reproducibility:** Figures, annotations, and supplementary materials have been refined to enhance interpretability. We also provided more precise explanations of model components, implementation details, and quantitative metrics.

We believe these revisions comprehensively address the reviewers’ major concerns regarding methodological clarity, empirical depth, robustness, and presentation quality. We remain grateful for the opportunity to improve our submission and are happy to clarify anything further.

Sincerely,
The Authors

---

### Meta-Review · Area_Chair_Evac · 2025-12-28

**Summary:**

This submission proposes **SeaVGGT**, a physics-guided self-supervised adaptation of VGGT for underwater depth estimation, using prototype-graph token modulation. Reviewers agree the problem is important and the approach is directionally interesting, with two reviewers leaning accept (Wnbj, 6HLZ:) and two reviewers leaning reject (pYbY, 95qr).

After considering the rebuttal and the authors’ revisions, the paper is improved in clarity and includes additional baselines/ablations. However, the overall evaluation remains below the acceptance threshold. The key reasons are: (i) the empirical evidence, while expanded, still leaves insufficient contextualization and qualitative substantiation of claimed gains across datasets/tasks; (ii) the paper’s presentation issues, though partially addressed, still make it hard to verify what exactly improves and why; and (iii) some components (e.g., prototype GNN choice and self-supervised proxy losses) remain under-justified relative to their complexity and potential instability. Therefore, the recommendation is **Reject**.

**Reviewer Concerns:**

### Issues that were addressed (partially) in the rebuttal

* **Broader baseline comparisons on FLSea and USOD10K**: authors added underwater pretrained baselines (UDepth/TRUDepth) and terrestrial pretrained models fine-tuned on underwater data (MDepth2/AdaBins), plus zero-shot foundation models (DAv2/VGGT/SeaVGGT), and provided extended tables. *(Reviewer pYbY)*

* **Clarifications of confusing presentation elements**: authors revised/relocated figures, clarified multi-view vs single-view rows, explained missing GT in reconstruction rows, and added bounding boxes and annotations. *(Reviewer pYbY)*

* **Hyperparameter sensitivity (loss weights)**: authors added a loss-weight sensitivity study and then provided explicit RMSE tables after the reviewer noted the figure lacked readable values. *(Reviewer 6HLZ)*

* **Clarification of “simplified imaging model”**: authors clarified $\beta$ is channel-wise (RGB) and provided extra ablations about background loss robustness to noise in depth proxy. *(Reviewer Wnbj)*

* **Prototype initialization / $\tau$ sensitivity**: authors provided ablations for random initialization and $\tau$ sweep. *(Reviewer Wnbj)*

* **GNN vs simpler alternatives**: authors added a justification and an ablation comparing graph attention vs NN assignment vs metric-learning baseline. *(Reviewer 95qr)*


### Core issues that remain and drive the Reject decision

* **Empirical strength and contextualization remain insufficient for the claimed impact**
  Even with additional tables, reviewer concerns about how compelling the gains are and what they mean in practice are not fully resolved. Several improvements are numerically modest, and the paper still struggles to clearly communicate where the model improves (foreground objects vs background water), under what conditions, and how robustly across datasets. *(Reviewers pYbY, 95qr; also echoed by 6HLZ noting improvements are “relatively marginal”)*

* **Qualitative evidence is still not fully convincing / hard to interpret**
  The rebuttal added new figures and annotations, but the fundamental critique remains: qualitative comparisons previously emphasized uninformative regions (e.g., water background), some datasets provide pseudo-depth GT of questionable reliability (USOD10K), and multi-view sections lack GT due to missing poses. These limitations are real, but they also make it harder to validate the claimed qualitative advantages, leaving the paper’s visual evidence less persuasive than needed for acceptance. *(Reviewer 95qr; also pYbY)*

* **Potential instability and dependence on proxy supervision is not fully settled**
  The background loss uses top-depth pixels as a proxy; authors provide noise injection experiments and show robustness up to moderate noise levels, but this remains a delicate self-training signal. The rebuttal demonstrates it “works in their setting,” yet questions about failure modes under more challenging optical conditions or poor initial depth remain. Given the limited availability/quality of underwater depth GT, this concern is hard to eliminate, but it still weakens confidence in general reliability. *(Reviewer Wnbj; to a lesser extent pYbY)*

**Reviewer Scores:**

* **Reviewer Wnbj (6)**: positive on efficiency and improvements; rebuttal strengthens their confidence, likely unchanged.
* **Reviewer 6HLZ (6)**: initially positive but noted the improvements are marginal; hyperparameter sensitivity visibility issues were addressed; likely unchanged.
* **Reviewer pYbY (4)**: major concerns on experimental explanation/context, missing comparisons, and confusing figures; rebuttal helps but likely not enough to change.
* **Reviewer 95qr (4)**: main concerns on qualitative evidence and justification of GNN/prototype design; rebuttal adds material but core skepticism likely remains.

---

### Decision · Program_Chairs · 2026-01-26

Reject